



# Development of physically based liquid water schemes for Greenland firn-densification models

Vincent Verjans[1], Amber Leeson[2], C. Max Stevens[3], Michael MacFerrin[4], Brice Noël[5] and Michiel R. van den Broeke[5]

5   [1] Lancaster Environment Centre, Lancaster University, Lancaster, LA1 4YW, UK.
[2] Lancaster Environment Centre / Data Science Institute, Lancaster University, Lancaster, LA1 4YW, UK
[3] Department of Earth and Space Sciences, University of Washington, Seattle, WA, USA
[4] Cooperative Institute for Research in Environmental Sciences, University of Colorado, Boulder, CO USA
[5] Institute for Marine and Atmospheric research Utrecht, Utrecht University, Utrecht, the Netherlands

*Correspondence to*: Vincent Verjans (v.verjans@lancaster.ac.uk)

**Abstract.**

As surface melt is increasing on the Greenland ice sheet (GrIS), quantifying the retention capacity of the firn layer is critical to link meltwater production to meltwater runoff. Firn-densification models have so far relied on empirical approaches to

account for the percolation-refreezing process, and more physically based representations of liquid water flow might therefore bring improvements to model performance. Here we implement three types of water percolation schemes into the Community Firn Model: the tipping bucket approach, the Richards Equation in a single-domain and the Richards Equation in a dual-domain, which accounts for partitioning between matrix and fast preferential flow. We investigate their impact on firn densification at four locations on the GrIS and compare model results with observations. We find that for all of the flow

schemes, significant discrepancies remain with respect to observed firn density, particularly the density variability in depth, and that inter-model differences are large. The simple bucket scheme is as efficient in replicating observed density profiles as the single-domain Richards Equation. The most physically detailed dual-domain scheme does not necessarily reach best agreement with observed data. However, we find that the implementation of preferential flow does allow for more frequent ice layer formation and for deeper percolation. We also find that the firn model is more sensitive to the choice of densification

scheme than to the choice of water percolation scheme. The disagreements with observations and the spread in model results demonstrate that progress towards an accurate description of water flow in firn is necessary. The numerous uncertainties surrounding firn micro- and macro-structure, its hydraulic properties, and the one dimensionality of firn models render the implementation of physically based percolation schemes difficult. An improved understanding of the parameters affecting evolution of polar firn, of the effects of the climatic forcing on the densification process and more accurate treatment of liquid

water would benefit further developments.



## 1 Introduction

Estimating the properties of the firn layer – and how it evolves under a warming climate – is a critical step in measuring the ice sheets' contribution to sea level rise, yet it remains one of the key sources of uncertainty in present assessments (McMillan et al., 2016). Accurate estimates of firn thickness and density are required for the conversion of space-borne measurements of volume change into mass change (e.g. McMillan et al., 2016; Shepherd et al., 2018). Also, assessments of the Greenland and Antarctic ice sheets' contribution to sea level require information on firn density and spatial distribution in order to calculate meltwater retention potential and the capacity of firn to buffer the flow of meltwater to the ocean (Harper et al., 2012; Machguth et al., 2016; van den Broeke et al., 2016). Surface melting has become more widespread and intense on the Greenland Ice Sheet (GrIS), with annual total melt rates rising by 11.4 Gt $yr^{-2}$ between 1991 and 2015 (van Angelen et al., 2014; van den Broeke et al., 2016). Most of this increase in melting has occurred in the percolation zone, where a firn layer is present year-round. This melt water percolates into the firn layer, where it can refreeze, run off, or remain liquid in temperate firn. Refreezing of liquid water in firn, known as internal accumulation, has an impact both on ice-sheet mass balance and on heat fluxes from the surface to the ice sheet (van Pelt et al., 2012). As such, understanding physical processes in firn, including in particular the transport of liquid water, is becoming increasingly important in order to accurately constrain and predict the mass balance of the GrIS (van den Broeke et al., 2016).

Densification of dry firn is typically modelled as a function of near-surface air temperature and accumulation (Herron and Langway, 1980; Arthern et al., 2010; Li and Zwally, 2011; Simonsen et al., 2013; Morris and Wingham, 2014; Kuipers Munneke et al., 2015a,b). If applied to wet firn, these models are often modified to include a simplified representation of liquid water percolation, the 'tipping bucket' scheme, which assumes flow and refreezing through the firn column occur in a single time step (Simonsen et al., 2013; Kuipers Munneke et al., 2015b; Steger et al., 2017). Observations have shown that, in actuality, liquid water transport in firn is characterised by flow patterns that are heterogeneous in space and time (Pfeffer and Humphrey, 1996; Humphrey et al., 2012). Incorporation of liquid water schemes representing such flow patterns would enable models to better represent the transport of mass and heat through the firn; these schemes might also improve modelled densification in wet firn conditions (Kuipers Munneke et al., 2014; van As et al., 2016). Liquid water flow however, is a complex function of several properties and processes that are difficult to constrain by observations and, as a corollary, are difficult to represent in these models (e.g. presence of impermeable ice layers, snow hydraulic properties, grain-size, lateral runoff). The infiltration of water through firn can be partitioned between the progressive advance of a uniform wetting front through the porosity, called matrix flow, and fast, localised, preferential flow (Waldner et al., 2004; Katsushima et al., 2013). This dual-nature of water flow has been reported in observations of the firn layer of the GrIS, where preferential flow pathways come in the form of discrete vertical conduits and are crucial to effectively transport surface meltwater in deep subfreezing firn (Pfeffer and Humphrey, 1996; Parry et al., 2007; Humphrey et al., 2012; Cox et al., 2015). The detection of Perennial Firn Aquifers (PFA), in which large amounts (140 Gt) of liquid water is stored year round in deep firn, further emphasises the

importance of firn hydrology on the GrIS mass balance (Forster et al., 2014; Koenig et al., 2014). Snow models have developed liquid water schemes based on the Richards Equation (RE) to simulate matrix flow (Hirashima et al., 2010; Wever et al., 2014; D'Amboise et al., 2017). The RE is a continuity equation describing water flow in unsaturated porous media and is widely used in hydrological models. Recently, a preferential flow scheme has been included in the SNOWPACK model to account

for heterogeneous percolation (Wever et al., 2016). Until now however, such developments have not been translated into firn-densification models.

In this study, we describe and compare liquid water schemes of different levels of physical complexity from snow models, and we apply these in combination with firn-densification models in order to evaluate the impact of the treatment of liquid water

flow on modelled firn densification and temperature. We use the Community Firn Model (CFM) as the modelling framework for our study; the CFM is able to simulate numerous physical processes in firn and includes a large choice of governing formulations for densification. We use the common tipping bucket approach and also develop schemes for liquid water flow in firn following physically based advances in snow models (Wever et al., 2014, 2016; D'Amboise et al., 2017).

We simulate liquid water flow and firn densification starting from 1980 at four sites on the GrIS: DYE-2, NASA-SE, KAN-U

and a PFA site (Fig. 1). These sites were chosen because they are collocated with recently drilled firn cores which allow a direct comparison of model results with observations. By comparing simulated firn densification to observations at these sites, we investigate the sensitivity of the system to the choice of liquid water flow scheme and the sensitivity of the flow schemes to various parameterisations of firn structural properties. Finally, we perform simulations with a range of firn-densification formulae and assess the relative importance of the choice of liquid water flow scheme to the choice of the underlying

densification equation.

## 2 Firn Model and Data

In this study we use and further develop the CFM, an open-source firn-densification modelling framework. We refer the reader to Stevens (2018) for details and briefly summarise the main characteristics here. The CFM is one-dimensional and works in

a Lagrangian framework; it is forced at its upper boundary by observed or modelled values for accumulation, surface temperature, surface density, rain, and snow melt. The CFM includes many of the commonly used dry firn-densification schemes (e.g. Herron and Langway, 1980; Helsen et al., 2008; Arthern et al., 2010; Li and Zwally, 2011; Morris and Wingham, 2014; Kuipers Munneke et al., 2015b). We refer the reader to the original publications for details on the different densification schemes and briefly outline the expressions used in our simulations in this section.




## 2.1 Dry firn-densification model

As a base case, we use the firn-densification formulation implemented in the snow model CROCUS (Vionnet et al., 2012), Eq. (1). It has previously been used in model studies of firn densification on the GrIS and also on polar ice caps (Gascon et al., 2014; Langen et al., 2017). This model is formulated so that densification is based on the overburden stress:

$$\frac{d\rho}{dt} = \rho \, \frac{\sigma}{\eta}$$
(1)

where $\rho$ is the density of the firn (kg m$^{-3}$), $\sigma$ is the stress due to weight of the upper layers (kg m$^{-1}$ s$^{-2}$) and $\eta$ is the snow viscosity (kg m$^{-1}$ s$^{-1}$) following the parameterisation:

$$\eta = f_1 f_2 \eta_0 \frac{\rho}{c_\eta} \exp\left[a_\eta (273.15 - T) + b_\eta \rho\right]$$
(2)

where $\eta_0 = 7.62237$ kg s$^{-1}$ m$^{-1}$, $a_\eta = 0.1$ K$^{-1}$, $b_\eta = 0.023$ m$^3$ kg$^{-1}$ and $T$ is the firn temperature (K). The parameter $c_\eta$ is set to
358 kg m$^{-3}$ as suggested by van Kampenhout et al. (2017) when using Eq. (1) for polar firn. There are two additional correction factors, $f_1$ and $f_2$, depending on firn microstructural properties. The factor $f_1$ accounts for the presence of liquid water:

$$f_1 = \frac{1}{1 + 60\,\theta}$$
(3)

where $\theta$ is the volumetric water content (m$^3$ m$^{-3}$). In this study, we neglect the change in snow viscosity for grain-sizes smaller than 0.3 mm by keeping the constant value $f_2 = 4$ (after Langen et al., 2017; van Kampenhout et al., 2017).

Several firn-densification equations have been derived and calibrated for the GrIS specifically. We favoured the use of Eq. (1) as our base case because (i) most of these calibrated schemes were developed for dry firn densification whereas the CROCUS formulation accounts for the presence of liquid water explicitly, (ii) applying a percolation scheme in a stress-based densification model rather than in an accumulation-rate-based model ensures that the redistribution of mass associated with percolation will affect the densification appropriately and (iii) the CROCUS densification scheme is currently used by the
regional climate model MAR and by the Earth System Model CESM to quantify firn densification on the GrIS (Fettweis et al., 2017; van Kampenhout et al., 2017).

## 2.2 Climatic forcing

To force the model at its upper boundary we use three-hourly skin temperature, melt, snowfall, rain and sublimation fields simulated by the latest version of the RACMO2 regional climate model (RACMO2.3p2, Noël et al., 2018). This model has a
5.5 km horizontal resolution grid and has been explicitly adapted for use over the polar ice sheets.

If the solid input rate (snowfall – sublimation) is negative over a time step, the CFM treats it by a corresponding mass loss for the surface layer and liquid water is evaporated before solid mass gets sublimated. The temperature of a newly accumulated snow layer is defined as the skin temperature at that time step. Deep temperatures in the model are thus mostly determined by the mean surface temperature applied during the spin-up process (Sect. 2.5) together with latent heat release through refreezing.
We use a Neumann boundary condition for the temperature at the bottom of the domain and use a 250 m deep column to account for the large thermal mass of the ice sheet during the transient run.





In addition to latent heat release due to refreezing, the CFM accounts for heat conduction through the different layers to determine the temperature profile. In accordance with previous firn modelling studies (Kuipers Munneke et al., 2015a; Steger et al., 2017), we make the firn conductivity, $k_s$, a function of density following Anderson (1976):

$$k_s = 0.021 + 2.5 \left(\frac{\rho}{1000}\right)^2 \tag{4}$$

Another boundary condition is the density of every fresh snow layer deposited at the surface. To reduce the sources of possible uncertainties, we simply use a constant and site-specific surface density according to the surface value of the drilled firn cores instead of a parameterised formulation.

**2.3 Grain-size**

The temporal evolution of grain-size in firn is poorly understood and observational constraints are scarce. However, the grain-
10 size is a key variable for the RE, and the flow schemes used in this study thus require an initial grain-size and a grain-growth rate. For the former, we use the empirical formulation of Linow et al. (2012) derived from observations of snow samples from Antarctica and Greenland:

$$r_0 = (b_0 + b_1(T_{av} - 273.15) + b_2 \dot{b} \frac{\rho_i}{\rho_w}) \tag{5}$$

Where $\rho_i$ is the ice density (917 kg m$^{-3}$), $\rho_w$ that of liquid water (1000 kg m$^{-3}$), $\dot{b}$ is the mean annual accumulation rate (m w.e.
15 yr$^{-1}$), $T_{av}$ the mean annual surface temperature (K) and $b_0$, $b_1$ and $b_2$ are calibration parameters taking the values 0.781, 0.0085 and -0.279 respectively.

For grain growth rate, the relationship proposed by Katsushima et al. (2009) is applied:

$$\frac{dr}{dt} = \frac{1}{8 \, r^2 \, 10^9} \, min \left[\frac{2}{\pi}\left(1.28 \, 10^{-8} + 4.22 \, 10^{-10} \left(\theta_{weight,\%}\right)^3\right), 6.94 10^{-8}\right] \tag{6}$$

where $r$ is the grain radius (m) and $\theta_{weight,\%}$ is the mass liquid water content expressed in percent and is thus related to $\theta$ (see
Eq. (3)):

$$\theta_{weight,\%} = 100 \, \frac{\theta \, \rho_w(\rho_i - \rho)}{\rho_i \, \rho} \tag{7}$$

Equation (6) combines a wet snow metamorphism formula and a higher limit of growth rate of ice particles, both derived from laboratory measurements.

To study the sensitivity of the model to the grain-size implementation, we also use an alternative option based on the approach
for West-Antarctic firn of Arthern et al. (2010): the grain radius in newly deposited layers ($r_0$) has the constant value of 0.1 mm and the grain growth rate is formulated as:

$$\frac{dr}{dt} = \frac{1}{2 \, r} \, k_g \, exp\left(\frac{-E_g}{R \, T}\right) \tag{8}$$

where $E_g$ is the activation energy for grain growth (42.4 kJ mol$^{-1}$), $R$ is the gas constant (8.314 J mol$^{-1}$ K$^{-1}$) and $k_g$ a parameter that takes the value 1.3 10$^{-7}$ m$^2$ s$^{-1}$. Note that Eq. (8) does not take the impact of liquid water presence on firn metamorphism
into account.





## 2.4 Study sites

We perform model simulations at four study sites of the percolation zone of the GrIS: NASA-SE, DYE-2, KAN-U and FA13 (perennial Firn Aquifer) (Fig. 1). The sites have different climatic conditions (Figs. 1 and 2) and well-documented firn cores are available in order to assess the performance of the different flow schemes. NASA-SE (66.48ºN, 42.50ºW, 2372 m a.s.l.) is located in the upper part of the percolation zone with a mean annual temperature of -20°C and relatively low melt rates (50 mm yr$^{-1}$). DYE-2 (66.48ºN, 46.28ºW, 2126 m a.s.l.) is a slightly warmer site ($T_{av}$ = -18°C), and melt is about three times greater than at NASA-SE (150 mm yr$^{-1}$). KAN-U (67.00ºN, 47.03ºW, 1838 m a.s.l.) is near the equilibrium line altitude and has warmer temperatures ($T_{av}$ = -8°C) and significant melting (280 mm yr$^{-1}$). FA13 (66.18ºN, 39.04ºW, 1563 m a.s.l., $T_{av}$ = -13°C) is a location where a firn aquifer has been recently discovered (Foster et al., 2014). The persistence of deep saturated layers year round is due to the coupling of high melt rates (587 mm w.e. yr$^{-1}$) with high accumulation rates (1002 mm w.e. yr$^{-1}$) (Kuipers Munneke et al., 2014). We perform transient firn-model simulations for each site until the date that a core was drilled. The cores at NASA-SE and DYE-2 were drilled in spring of the years 2016 and 2017 respectively, as part of the FirnCover project. The cores at KAN-U (Machguth et al., 2016) and at FA13 (Koenig et al., 2014) were drilled in spring 2013.

## 2.5 Spin-up and domain definition

In order to simulate the evolution of the firn layer in time, we start the transient simulation from an initial state in equilibrium with a reference climate. In accordance with previous GrIS firn studies (Kuipers Munneke et al., 2015b; Steger et al., 2017), we take the 1960-1979 climate as reference climate because it predates the onset of the general warming of Greenland and the subsequent increase in surface melt. We iterate over the reference climate until 70 m w.e. of snow has been accumulated, which ensures the entire firn column is refreshed. The number of iterations over the reference climate is thus site-specific. This spin-up process starts from an analytical solution for the density profile (Herron and Langway, 1980) with temperatures corrected to account for latent heat release by refreezing (Reeh, 2008). During the spin-up process we use the simple percolation bucket approach, and the more advanced flow schemes, detailed in the next section, are turned on only at the end of the spin-up for the transient simulation. This is because using the advanced schemes over long periods is computationally expensive. The domain on which the flow calculations are applied is a subset of the entire CFM domain; this sub-domain is defined each time the flow routine is called in the transient run. The bottom of the sub-domain is defined as the depth from which firn density does not reach values below the pore close-off value (830 kg m$^{-3}$), because infiltration of liquid water becomes negligible at this point. The thickness of the layers deposited in every three hourly time step determines the vertical resolution, and we apply a merging process only to individual layers less than 2 cm thick (see Appendix A8).

## 3 Liquid water schemes

The water flow schemes are added to the dry-densification model detailed in Sect. 2.1 and are thus also effectively one-dimensional, representing no lateral exchange of heat and mass although lateral runoff is used as a mass sink. In this section,



we present the three different flow schemes that we implement in the CFM: (1) the Bucket method (BK), (2) a single-domain Richards Equation scheme (R1M) and (3) a dual-permeability Richards Equation scheme (DPM). Because of its robustness and ease of implementation, BK is the current 'state-of-the-art' in firn-densifcation models that are interactively coupled to regional climate models. R1M is used in several stand-alone snow models to describe water flow (Hirashima et al., 2010; Wever et al., 2014; D'Amboise et al., 2017), and DPM is entirely based on the scheme implemented in the snow model SNOWPACK (Wever et al., 2016), where dual-permeability means that separate domains for matrix flow and preferential flow coexist with liquid water exchanged between these domains.

### 3.1 Bucket model

The 'tipping bucket' percolation scheme is commonly used to account for the vertical transport of meltwater in firn models, though the precise form of its implementation is variable. Each layer in the model can refreeze meltwater according to its 'cold content', i.e. the energy required to raise the temperature of the layer to the melting point. Starting from the surface, the meltwater may percolate through successive layers, thus allowing for refreezing at depth. Meltwater is progressively depleted due to refreezing and retention according to each layers' water-holding capacity, which is the part of the water that is stored in some of the available pore space and not subject to vertical transfer. The water-holding capacity acts as an approximation of the effect of capillary forces on water retention. Percolation proceeds until all the meltwater is stored (refrozen or retained) or until it reaches a layer with a density exceeding the impermeability threshold (780-830 kg m$^{-3}$), at which point lateral runoff is assumed. The BK thus requires two parameters: the water-holding capacity and the impermeability threshold. We test two possibilities for the former and three for the latter. The water-holding capacity can be prescribed by the calculations of Coléou and Lesaffre (1998) for the mass proportion of water in a firn layer, $W_w$:

$$W_w = 0.057 \frac{\rho_i - \rho}{\rho} \tag{9}$$

This mass proportion is then converted to the water-holding capacity, $\theta_h$:

$$\theta_h = \frac{W_w}{(1-W_w)} \frac{\rho \, \rho_i}{\rho_w (\rho_i - \rho)} \tag{10}$$

Using constant values of the water-holding capacity is also common practice (Reijmer et al., 2012; Steger et al., 2017). Our base case scenario uses a fixed $\theta_h$ at 0.02, or 2% of the pore space available for liquid water retention. This low value assumes effective downward percolation and is meant to account for vertical preferential flow (Reijmer et al., 2012). For that reason, we consider this as a good basis for comparison with the DPM that explicitly accounts for such vertical preferential flow.

We test three values for the impermeability threshold; these were selected in accordance with Gregory et al. (2014), who tested firn permeability of Antarctic samples in a lab and reported that impermeability can occur over density values ranging from 780 kg m$^{-3}$ to 840 kg m$^{-3}$. We thus take our three test values to be 780 kg m$^{-3}$, 810 kg m$^{-3}$ and 830 kg m$^{-3}$, respectively the lower bound and middle of this range and a commonly-used value of pore close-off density.



## 3.2 Richards Equation

Vertical movement of water in a variably saturated porous medium can be described by the one-dimensional version of the RE:

$$\frac{\partial \theta}{\partial t} - \frac{\partial}{\partial z}\left[K(\theta)\left(\frac{\partial h}{\partial z} + 1\right)\right] = 0 \tag{11}$$

where $K$ is the hydraulic conductivity (m s⁻¹), $h$ is the pressure head (m) and $z$ is the vertical coordinate (m, taken positive downwards). The +1 term accounts for the effect of gravity. The RE is an equation expressing the mass conservation law and Darcy's law and it includes the 'suction head', i.e. the suction force exerted at the surface of individual grains.

A water-retention curve describes the relationship between $\theta$ and $h$ required by Eq. (11). We use the van Genuchten (1980) model which is typically applied in studies of liquid water flow through snow (Jordan, 1995; Hirashima et al., 2014; Wever et

al., 2014; D'Amboise et al., 2017):

$$\theta = \theta_r + (\theta_{sat} - \theta_r)\frac{(1+(\alpha|h|^n)^{-m}}{Sc} \tag{12}$$

where $\theta_r$ is the residual water content (m³ m⁻³), $\theta_{sat}$ is the volumetric liquid water content at saturation (m³ m⁻³). $Sc$ is a correction coefficient following Wever et al. (2014). The parameters $\alpha$, $n$ and $m$ are fit coefficients, with $\alpha$ being related to the maximum pore size and $n$ and $m$ being related to the pore size distribution. These three parameters, referred to as the van

Genuchten parameters, are specific to the modelled porous medium and for snow; a common approach is to use the parameterisation developed by Yamaguchi et al. (2012) in a laboratory study:

$$\alpha = 4.4 * 10^6 \left(\frac{\rho}{2r}\right)^{-0.98} \tag{13}$$

$$n = 1 + 2.7 * 10^{-3}\left(\frac{\rho}{2r}\right)^{0.61} \tag{14}$$

$$m = 1 - \frac{1}{n} \tag{15}$$

Yamaguchi et al. (2012) measured the water-retention curve for a range of grain radii (0.025 to 2.9 mm) and densities (361 to 636 kg m⁻³) in different snow samples by using a gravity drainage column method.

The porosity is the part of the volume not occupied by the solid matrix and, in the case of firn, is defined as:

$$P = 1 - \frac{\rho}{\rho_i} \tag{16}$$

The volumetric liquid water content at saturation is proportional to the porosity (Wever et al., 2014):

$$\theta_{sat} = P\,\frac{\rho_i}{\rho_w} \tag{17}$$

Note that water is not assumed to fill the entire pore space in saturated conditions and the correction factor included in Eq. (17) accounts for the required space to allow the liquid water to freeze.

The parameter $\theta_{sat}$ thus represents the pore space available for liquid water and from there we can define the effective saturation as:

$$Se = \frac{\theta - \theta_r}{\theta_{sat} - \theta_r} \tag{18}$$




and $Se$ must be bounded between 0 and 1. In completely dry layers, a zero effective saturation would lead to infinite values in the head pressure calculation and thus, we use a numerical adjustment to avoid this happening (see Appendix A3). The residual water content $\theta_r$ is defined as the amount of liquid water that cannot be removed by gravity as it is held by capillary suction at the surface of the solid grains. Following Yamaguchi et al. (2010), a constant value of $\theta_r = 0.02$ can be taken but in case of

5 refreezing, $\theta$ can approach zero and $\theta_r$ must be adjusted accordingly. We take $\theta_r$ following a piecewise function:

$$\theta_r = \min[0.02, 0.9\,\theta] \tag{19}$$

Another issue with the numerical requirement of an effective saturation value strictly superior to zero is that very low flow rates persist, even for liquid water contents close to the residual water content. Over long time periods, layers cannot hold any residual water content and eventually dry out under the effect of gravity, contrarily to BK. By taking the coefficient 0.9 in Eq.

(19) instead of 0.75 used in snow models (Wever et al., 2014; D'Amboise et al., 2017), we partially reduce this effect.

The hydraulic conductivity ($K(\theta)$) is the ability of the fluid to flow through the porous medium under a certain hydraulic gradient that depends on pressure head and gravity. Thus, $K(\theta)$ depends on the effective saturation and on the properties of both the porous medium and the fluid; fluid flow is enhanced in highly saturated layers. The hydraulic conductivity is described by the van Genuchten-Mualem model (Mualem, 1976; van Genuchten, 1980):

$$K(\theta) = Ksat\,Se^{1/2}\left[1 - \left(1 - Se^{\frac{1}{m}}\right)^m\right]^2 \tag{20}$$

where $Ksat$ is the hydraulic conductivity in saturated conditions ($Se = 1$). For the case of water flow through snow, it has been inferred using three-dimensional images of the microstructure by Calonne et al. (2012) as:

$$Ksat = 3.0\,r^2\,\exp(-0.013\,\rho)\left(\frac{g\,\rho_w}{\mu}\right) \tag{21}$$

Where $g$ is the gravitational acceleration (9.8 m s$^{-2}$) and $\mu$ is 0.001792 kg m$^{-1}$ s$^{-1}$, the dynamic viscosity of liquid water at

20 273.15 K. Equation (21) shows that simulated water flow is faster in layers with coarser grains and lower densities. These conditions correspond to cases where the connectivity between the pore spaces is high. With respect to the hydraulic conductivity parameterisation, we additionally modify the permeability of ice layers. The hydraulic conductivity of any layer above the impermeability threshold is set to zero, rendering it impermeable to incoming flow and leading to the ponding of water on top of it. This RE implementation completely describes R1M and provides the basis of DPM, further detailed in the

25 next section.

Details of the numerical implementations that are required to maintain stability and to improve computational efficiency for the RE calculations are discussed in the Appendix.

### 3.3 Dual-permeability model

Physical models of preferential flow in snow are still scarce (Hirashima et al., 2014; Wever et al., 2016). In this section, we

explain how the SNOWPACK dual-permeability model (Wever et al., 2016) is implemented in the CFM. The firn column is separated into two domains; water flow in both is governed by the RE described in Sect. 3.2. We define $F$ as the pore space





allocated to the preferential flow domain and accordingly 1-$F$ as the pore space for the matrix flow domain. Wever et al. (2016) used a grain-size dependence for $F$, but their regression was performed on only four data points measured in idealised snow laboratory conditions (Katsushima et al., 2013). The experimental grain-sizes ranged from 0.1 to 0.8 mm and the water input from 480 to 550 mm per day, which is not representative of firn conditions in Greenland (Figs. 1 and 2). Moreover, due to the

typical grain-size ranges in firn (Gow et al., 2004; Lyapustin et al., 2009), the model would regularly be forced to use for $F$ the minimal value for numerical stability implemented in SNOWPACK. To deal with this uncertain parameter but still remain as close as possible to the SNOWPACK implementation, we favour the use of a constant value based on observations in natural snow. Marsh and Woo (1984) and Williams et al. (2010) reported that rapid flow paths occupy respectively 22% and 5% to 30% of the area and we thus fix the value $F = 0.2$. However, the extension of the preferential flow area within the snowpack

is very likely to be a function of grain-size and meltwater influx, but even in laboratory conditions these dependencies must still be investigated further (Avanzi et al., 2016). The value of $F$ thus determines the value of the saturated liquid water content $\theta_{sat}$ in both domains and instead of Eq. (17), we write:

$$\begin{cases} \theta_{sat,m} = (1-F)P\frac{\rho_i}{\rho_w} \\ \quad \theta_{sat,p} = F\,P\frac{\rho_i}{\rho_w} \end{cases} \tag{22}$$

where from now on, the subscripts $m$ and $p$ stand for matrix and preferential flow domain respectively. Equation (22) shows

that the volumetric water content in the preferential flow domain is smaller than that in the matrix flow domain. All the input of meltwater is added to the matrix flow domain. For the regulations of the exchanges of water between both domains, we also closely follow the transfer processes of SNOWPACK (Wever et al., 2016) which are executed at the same 15-minute time step. We briefly summarize the transfer processes below.

Water from the matrix flow domain can enter the preferential flow domain of the layer below if the pressure head in the layer

reaches the water entry suction, $h_{we}$, of the underlying layer. The parameter can be expressed as (Katsushima et al., 2013; Hirashima et al., 2014; Wever et al., 2016):

$$h_{we} = 0.0437(2r)^{-1} + 0.01074 \tag{23}$$

The amount of water transferred into the preferential flow domain equals the amount of water in excess of $h_{we}$. If after the transfer, $Se$ in the matrix flow domain still exceeds $Se$ in the preferential flow domain of the underlying layer, their respective

$Se$ are equalised by transferring the appropriate amount of water from the overlying matrix flow domain to the underlying preferential flow domain. In addition, in every individual firn layer where $Se$ in the matrix flow domain exceeds $Se$ in the preferential flow domain, matrix and preferential $Se$ are equalised by transferring water from the matrix flow domain to the preferential flow domain. This serves to avoid the presence of horizontal pressure gradients in wet snow.

Water can flow from the preferential flow domain to the matrix domain by two processes. The first process is when the

saturation in the preferential flow domain exceeds a threshold value Θ. Wever et al. (2016) determined Θ by tuning its value to best match observations. When this threshold is reached, the amount of water corresponding to the cold content of the layer flows back into the matrix domain. If there is still water in excess of the threshold in the preferential flow domain, saturation



in both domains is set equal to one another. The second process simulates the heat flow from the preferential flow domain (at the melting point) to the colder surrounding matrix domain. Instead of transferring sensible heat, this process allows liquid water and its inherent latent heat to be exchanged to account for a theoretical heat flow, $Q$, and thus approximating Fourier's law:

$$Q = k_s \frac{(T - 273.15)}{\left( \sqrt{\frac{1+F}{2\pi}} - \sqrt{\frac{F}{\pi}} \right)} \tag{24}$$

This formulation assumes a linear horizontal temperature gradient in the matrix and a circular shape of the preferential flow path's perimeter. From Eq. (24), the corresponding water transfer is calculated as:

$$\Delta\theta_{p \to m} = \frac{2 \, N \, \sqrt{\pi F} \, Q \, \Delta t_{15}}{L_f \, \rho_w} \tag{25}$$

where $\Delta t_{15}$ is the 15 minutes time step (s), $L_f$ is the specific latent heat of fusion (335 500 J kg$^{-1}$) and $N$ is a tuning parameter
representing the number of preferential flow paths per square meter (m$^{-2}$). In their study, Wever et al. (2016) arrived at a best parameter set for $\Theta$ and $N$ of 0.1 and 0 m$^{-2}$ based on comparisons of ice-layer occurrence and runoff amounts with observations in alpine snowpacks. Note that the use of a null value for $N$ is implausible in our case of firn-column simulations. Indeed, this would imply that liquid water would persist and flow deeper in the preferential flow domain in saturation conditions below the $\Theta$ value until the bottom of a subfreezing firn column, which can be up to 70 meters thick in some areas of the GrIS. Therefore,
we use the smallest non-zero value of $N$ tested by Wever et al. (2016) and the parameters $\Theta$ and $N$ are fixed to 0.1 and 0.2 m$^{-2}$ respectively.

The hydraulic conductivity of ice layers is not synthetically set to zero in the preferential flow domain as it is in the matrix flow domain. Preferential flow thus provides a way for water to flow through an ice layer, reproducing observations that ice layers are not totally impermeable barriers and can lead to localised piping events (Marsh and Woo, 1984; Pfeffer and
20 Humphrey, 1998; Williams et al., 2010; Sommers et al., 2017). An exception for this is the bottom of the domain: as preferential flow is stopped at the last layer, it does not percolate through the surface of the ice sheet.

### 3.4 Additional processes in the single- and dual-domain schemes

### 3.4.1 Refreezing process

In R1M and DPM a 'cold content' is calculated for every firn layer, similarly to BK (Sect. 3.1) and refreezing is executed at
25 the same frequency as the water transfer processes of DPM.

When refreezing occurs, every layer freezes the maximum of its liquid water content that its cold content allows. For numerical reasons, refreezing cannot dry out a layer completely; instead, a very low value of liquid water remains in every layer (see Appendix A3). The refrozen water densifies the firn layer and modifies its hydraulic properties. The remaining liquid water is still subject to flow and infiltrates deeper into the firn column.
In DPM, refreezing is restricted to the matrix flow domain (see Appendix A7). In the preferential flow domain, liquid water can percolate through cold layers, as this has been observed in field studies on the GrIS (e.g. Pfeffer and Humphrey, 1996;





Humphrey et al., 2012). For this liquid water to refreeze, it first has to be transferred back to the matrix flow domain. Preferential flow thus provides a way for liquid water to bypass cold firn layers and subsequently to infiltrate deeper layers.

### 3.4.2 Aquifer development and lateral runoff

In SNOWPACK, all the water reaching the bottom of the snow column is assumed to run off. In R1M and DPM, we allow for ponding of water at the bottom of the firn column (on the top of the solid ice surface) to enable the progressive formation of firn aquifers that exist on the GrIS (Forster et al., 2014; Kuipers Munneke et al., 2014). The model identifies the layers on top of the ice sheet that are saturated with meltwater and does not perform the flow calculations in this lowest part of the domain (see Appendix A5). All the inflow of water reaching this section is added to the aquifer, hence allowing the model to progressively fill the pore space of the bottom layers in the firn column with meltwater.

Despite the conservation of the water reaching the bottom layers in the domain, lateral runoff is still implemented in the rest of the column and is simulated by using the parameterised formulation of Zuo and Oerlemans (1996):

$$\frac{dRu}{dt} = \frac{L_{excess}}{\tau_{Ru}} \tag{26}$$

$$\tau_{Ru} = c_1 + c_2 \exp(-c_3\, S) \tag{27}$$

where $Ru$ is the amount of meltwater that runs off (m), $L_{excess}$ is the excess of liquid water amount with respect to the residual water content (m) and $\tau_{Ru}$ is a characteristic runoff time (s). The constants $c_1$, $c_2$ and $c_3$ are parameters derived by Zuo and Oerlemans (1996) for the GrIS and $S$ is the surface slope. The meltwater input is immediately treated as lateral runoff if the surface layer is an impermeable ice layer or if it is saturated.

Equation (26) leads to the complete drainage of a layer with a zero slope in only 26 days, which precludes the formation and persistence of perennial firn aquifers. Therefore, runoff is not applied in the layers at the bottom of the firn column where a firn aquifer is building up. The water in such aquifers in the lowest layers has been demonstrated to be ponding over long time periods (Forster et al., 2014) and is affected by other drainage processes not represented in the model such as entering crevasses (Poinar et al., 2017) and possibly catastrophic water release events (Koenig et al., 2014). However, not applying any runoff could theoretically lead to an infinite liquid water accumulation until the water table reaches the surface of the firn layer and we use a pragmatic approach to solve this issue. At the firn aquifer site, Koenig et al. (2014) measured a total water mass of 18.7 kg in 12 cm diameter boreholes, which thus corresponds to 1.65 m w.e. Because of the dearth of data indicating how much water might be stored in PFAs and the difficulty of accounting for horizontal drainage processes in a one-dimensional model, we use this value as a model threshold: any amount of water in excess of this value becomes runoff. In firn aquifers forming at the bottom of the firn column, the saturation in both domains is equalised.

### 3.5 Investigating model sensitivity

In Sections 2 and 3, we highlight several factors influencing BK, R1M and DPM. For each of the schemes, we analyse results generated using three possible impermeability thresholds: 780 kg m$^{-3}$ (ip780), 810 kg m$^{-3}$ (ip810) and 830 kg m$^{-3}$ (ip830). This





provides a way to compare the sensitivity of the simple BK and of the physically based schemes (R1M and DPM) to a common parameter. For BK, we try two different formulations of the water-holding capacity: constant at 0.02 (wh02) and according to the parameterisation of Coléou and Lesaffre (1998), Eq. (9) (whCL). For R1M and for DPM, we test two different grain-size implementations: the Linow et al. (2012) surface grain-size calculation, Eq. (5) coupled to the Katsushima et al. (2009) grain

growth rate, Eq. (6) (grLK) and the grain-size implementation of Arthern et al. (2010), Eq. (8) (grA). It is important to examine model sensitivity to the grain-size variable as almost all the hydraulic parameters of the RE depend on it. The different sensitivity tests are summarised in Table 1.

## 4 Results

In this section, we describe and discuss the model performances at each of the four sites tested (DYE-2, NASA-SE, KAN-U

and FA13). We systematically start by comparing BK, R1M and DPM in a base case parameterisation: BK wh02 ip810, R1M grLK ip810 and DPM grLK ip810 respectively. Then, we proceed to various tests to investigate the sensitivity of the flow schemes to variations in their parameter values. We refer to ice layers as layers with a density value exceeding the impermeability threshold in the model, and to liquid water input as the total of meltwater and rain influx. The DPM approach features two tuning parameters, $N$ and $\Theta$. Model results and depth-density profiles were found to be weakly sensitive to the

value of $N$ and $\Theta$ and so we omit consideration of these from the remainder of our study. The firn air content (FAC) is the depth integrated porosity in a firn column. We introduce this quantity because we make use of it to compare results of simulations with each other and with observations. We systematically quantify FAC over the first 15 m of depth.

### 4.1 DYE-2

DYE-2 has a typical liquid water input between 0.1 and 0.3 m w.e. yr$^{-1}$ (Fig. 2), which is moderate in the context of our study

sites. The extreme melt year of 2012 (Nghiem et al., 2012) is an exception, with an estimated input of more than 0.7 m w.e. Using BK, almost all of this meltwater refreezes locally and runoff is close to zero (Table 2) until the 2012 summer when ice layers ($\rho \geq 810$ kg m$^{-3}$) start forming in the top 2 m (Fig. 3a). Runoff increases in the subsequent years because meltwater reaches these ice layers. In R1M and DPM, small amounts of runoff occur between 1980 and 2011 due to the lateral runoff implementation, Eq. (26). Beginning in summer 2004, some ice layers start to form in R1M (Fig. 3b) due to the refreezing of

water held close to the surface by capillary forces. Over the 2012 summer, surface layers are progressively melted, bringing ice layers closer to the surface. The ponding and refreezing of water on the top ice layer allows it to thicken. This then acts as an impermeable barrier to vertical percolation from 2012 onwards, resulting in a more than sixfold increase in runoff (Table 2). In contrast, runoff remains low in DPM, in which several ice layers form in the upper firn as early as summer 1996 (Fig. 3c). These ice layers generally form deeper than 2 m due to more effective water transfer from the near-surface to lower layers;

preferential flow provides a path for ponding meltwater in the matrix flow domain to bypass ice layers and continue to percolate vertically, thus maintaining low runoff amounts. Preferential flow brings part of the 2012 meltwater to depths greater than 12



m. For each flow scheme, the modelled FAC underestimates the observed value by 4-16%. This can partly be attributed to the tendency of the CROCUS scheme to slightly overestimate densification rates in the upper part of polar firn (Gascon et al., 2014). FAC is underestimated more strongly in DPM (16 %) than in BK and R1M (4 %) because in DPM the deeper firn is not isolated from surface meltwater percolation (Table 2).

Modelled density profiles using each flow scheme are compared with observations (Fig. 4a). Mean density is reproduced reasonably well with each of the three flow schemes, but no configuration is able to qualitatively reproduce the strong variability in density observed. For example, numerous high-density layers separated by much lower density intervals are clear in the observations. Regardless of the flow scheme, only a few ice layers are formed in the model and these tend to be confined
to the upper 6 m, which has been affected by the higher melting rates of the recent years. In older firn deposited under lower-melt conditions, the number of density peaks and their amplitude is underestimated even more strongly. Several ice layers are observed in the 10 – 20 m depth range where only DPM simulates the presence of ice layers.

The three flow schemes lead to significantly different firn thermal conditions. The temperatures at 10 m depth of BK and R1M
agree well with observations (+0.2 and -0.4 K). In contrast, 10 m temperature is strongly overestimated in DPM (+2.7 K) because it allows percolation at depth, subsequent refreezing, and latent heat release. The summer 2012 percolation raises the 10 m depth temperature to within a few degrees of melting using DPM. Since the DPM method seems to exaggerate deep percolation, we tested a lower impermeability threshold (DPM grLK ip780) which should favour the formation of shallow ice layers, the ponding of water in the matrix flow domain, more lateral runoff and colder temperatures at depth. The ice layers do
form slightly earlier in the melt seasons but not noticeably shallower than in DPM ip810. The partitioning between runoff and refreezing is barely affected and the 10 m temperature bias remains (Table 2).

The BK method gives a density profile closer to R1M than to DPM. In order to mimic the behaviour of DPM we increase the impermeability threshold in BK (BK wh02 ip830) to make it more effective in transporting water vertically; however, model
results are only weakly affected by this change (Table 2). We also modify the water-holding capacity in BK according to the parameterisation of Coléou and Lesaffre (1998) (BK whCL ip810) which allows more water to be retained in the low-density layers close to the surface. Ice layers appear earlier in the simulation and at shallower depths (Fig. 3d). This increases the amount of runoff in BK whCL ip810 with respect to BK wh02 ip810 (+4 % of the water input over the entirety of the transient model run); however, in the surface layers, where high amounts of water are retained, refreezing dominates. As a result, much
less water percolates to the deeper firn and there is less refreezing and latent heat release. All of this leads to a significantly higher FAC (+4 %) and colder 10 m temperature (-1.5 K).

For models based on the RE (R1M and DPM), we test sensitivity to grain-size by implementing a parameterisation for grain growth based on Arthern et al. (2010) (experiments grA). Using this parameterisation, grain-sizes tend to be smaller, and so



more water tends to be retained and refrozen close to the surface due to stronger capillary forces. Compared to the R1M grLK ip810 experiment, the R1M grA ip810 causes formation of ice layers earlier in the simulation (beginning in 1996) and shallower in the firn column (Fig. 3e), favouring water ponding and subsequent runoff (+7 % of the water input over the entirety of the transient model run). Stronger capillarity also means that saturation is higher for percolation to occur, which in turn increases the simulated runoff since more water is in excess of the residual water content. The enhanced runoff and shallower percolation lead to a higher FAC (+4 %) and a colder 10 m temperature (-0.6 K). In DPM, the flow and refreezing patterns are also altered by the grain-size formulation: DPM grA ip810 produces ice layers much earlier (beginning in summer 1981), at shallower depths and in larger numbers (Fig. 3f). Runoff is however only slightly increased (+2%). The FAC remains similar to DPM grLK ip810, but the 10 m temperature is 0.3 K lower  and the warm bias is thus reduced (an 11 % decrease) (Table 2).

Finally, we investigate differences in the depth-density profiles simulated at DYE-2 attributed to different firn-densification formulations in contrast to those observed due to the use of different flow schemes. We first choose to apply the DPM grLK ip810 flow scheme with the additional firn-densification formulations of Herron and Langway (1980) (HL) and of Kuipers Munneke et al. (2015) (KM), both calibrated for GrIS firn. The FAC (-5 %), 10 m temperature (+0.4 K) and mean density profile (Fig. 4b) predicted by the HL densification model agree reasonably well with the CROCUS model, although HL predicts greater density variability due to its stronger dependence on the annual temperature cycle. In contrast, the KM model predicts much higher densification rates and thus greater densities, with several thick ice layers in the 3-8 m depth range, some exceeding a meter thickness. This results in a much lower FAC value compared to the CROCUS model (-24 %) and in this case, differences between flow schemes are small with respect to the choice of the densification formulation. Since the warm bias of DPM can cause temperature-dependent densification formulations to overestimate densities, we also compare the three densification formulations coupled to R1M grLK ip810 (Fig. 4c). Similar to the results using the DPM flow scheme, the HL profile agrees reasonably well with the CROCUS model (FAC value is -7 %) but predicts that a meter-thick ice layer formed at 5 m depth (Fig. 4c) during the 2012 summer. Discrepancies between CROCUS and KM are only slightly reduced using R1M; for example, the FAC predicted by KM is 20% less than that predicted by CROCUS. This can be attributed to greater densities at depth (>8 m) and to much higher densities in the depth range 3-5 m. The latter corresponds to the layers affected by meltwater refreezing and considerable latent heat release in the 2012 summer.

## 4.2 NASA-SE

NASA-SE is a site characterised by high accumulation rates, ranging between 0.5 and 0.8 m w.e. yr[-1], and low rates of liquid water input, typically between 0.01 and 0.15 m w.e. yr[-1] (Fig. 2). Under these conditions, abundant pore space and cold content are available for prompt refreezing of the summer meltwater so one expects a smaller sensitivity of the model to the flow scheme applied. In BK, no runoff is produced over the entire simulation (Table 3) since refreezing of small amounts of melt does not lead to the formation of impermeable ice layers. R1M and DPM have very low runoff amounts with a small spike in





the summer of 2012 when there was 0.38 m w.e. of liquid water input. No ice layer forms in the top 15 m of the firn column using any of the liquid water schemes, in agreement with the observed core (Fig. 5a). Changing the impermeability threshold results in identical model results since no layer exceeds the lowest possible value in the depth range where water percolates. The three water-transport schemes predict a similar FAC; they all underestimate the observed value by approximately 3%

(Table 3). This is because the mean firn density is well-captured by the model but somewhat overestimated in the lower part of the core (Fig. 5a). R1M simulates a single density peak at 8 m depth (Fig. 5a), corresponding to the 2012 summer meltwater percolation, due to capillary forces effectively retaining the relatively high meltwater volume produced in that year close to the surface and exposing it to delayed refreezing once these layers cool below the freezing point. DPM also produces a density peak (albeit a much smaller one) at a similar depth, and more-effective downward percolation results in a uniform increase in

density over the next 3 m. Finally, BK also produces a small density peak; however, this is at a greater depth of 9 m since it assumes water flow to be instantaneous in a time step and the major part of the refreezing occurs as water reaches deeper cold layers. Again, none of the percolation scheme captures the observed variability in density. Also, despite the low melt/accumulation ratio, the three percolation schemes overestimate the 10 m temperature by 1.4-2.2 K (Table 3).

Increasing the water-holding capacity in BK (BK whCL ip810) leads to a minor increase in the FAC (< 1%) and a 0.9 K cooling of the 10 m temperature, because the surface layers have a relatively low density (surface boundary condition of 240 kg m$^{-3}$ at this site) and thus retain high amounts of water with the CL parameterisation (Table 3). The R1M and the DPM density profiles are weakly sensitive to a change in the grain-size formulation to grA (Table 3). This is due to the small meltwater amounts with meltwater refreezing only slightly closer to the surface because of the stronger capillarity retention in

the grA models. However, it is noteworthy that simply changing the grain-size formulation in R1M leads to a 0.4 K colder 10 m temperature and thus decreases the bias with respect to observations by 28 % (Table 3).

We used the R1M grLK ip810 model with the HL and the KM densification formulations in order to prevent the DPM's warm bias from skewing the modelled densification rates. As expected in this relatively dry site, the modelled profiles are much

more sensitive to the dry-densification than to the percolation scheme (Table 3 and Fig. 5a and b). The aim of this paper is not to discuss the specificities of the dry-densification schemes, but the maximal difference in FAC among the three densification formulations tested is 20% compared to less than 1 % among the three flow schemes and their possible parameterisations. In contrast with the DYE-2 simulations, the CROCUS model predicts the fastest densification and thus the lowest FAC. HL and KM predict 20% and 11% greater FAC than CROCUS, respectively, and CROCUS is in closest agreement with the

observations.

## 4.3 KAN-U

KAN-U is a high-melt site with an average melt rate over the 1980-2013 period of 0.33 m w.e. yr$^{-1}$, and in the last three years of our simulation (2010-2013), the RCM calculates annual melt exceeding annual accumulation (Fig. 2). Since surface





temperatures are relatively high (annual mean around -8 °C), refreezing of the summer meltwater depletes the cold content over large depth ranges. Beginning in summer 1990 in the BK simulation, some ice layers are present in the depth range 3-8 m (Fig. 6a), allowing part of the meltwater to runoff and impeding percolation to greater depths. At the start of 2012, there is a thick ice layer in the upper 4 m and another one forms at the surface during the summer. As a result, refreezing is constrained

to the uppermost firn layers and a large part of the water input runs off (Table 4). In R1M, the high water content and the almost-continuous presence of ice layers in the upper 5 m from summer 1986 onwards (Fig. 6b) cause relatively high runoff rates throughout the simulation (28 % of the water input over the entirety of the transient model run). As in the BK simulation, runoff is particularly high in 2012 due to ice layers impeding vertical percolation below 1 m (Table 4). In the DPM simulation, the preferential flow mechanism leads to the formation of multiple ice layers in the depth range 4-10 m from 1987 onwards

(Fig. 6c). Runoff rates remain low but there is a notable increase in 2012. This is due to the formation of ice layers close to the surface, which allows ponding of water in the matrix flow domain. The preferential flow domain is unable to accommodate all the ponding water, and part of it is treated as lateral runoff (Eq. (26)). While matrix flow typically remains constrained to the upper 5 m (Fig. 7a), the recent (2010 to 2012) high-melt summers cause preferential flow to reach much greater depths (e.g. up to 35 m in the 2012 summer (Fig. 7b)). Since preferential flow can transfer water below ice layers, the refreezing

process can fill the pore space available at depth, leading to substantial thickening of the ice layers. As a result, the FAC is much smaller in the DPM simulation than in the BK (-39 %) and the R1M (-35 %) simulations.

The observations reveal a thick, almost continuous ice slab over the depth range of 1-7 m (Fig. 8a). Below it, the density is more variable but remains generally high causing a low FAC (Table 4). Both the BK and the R1M simulation significantly

overestimate the FAC (+59% and +50 %) whereas the DPM simulation agrees very well with the observed value (-2 %). However, the DPM density profile shows an almost continuous ice slab from 3 to 17 m depth (Fig. 8a) and does not reproduce the lower density intervals observed. This demonstrates an important limitation of the liquid water schemes: since water cannot be retained in layers exceeding the impermeability threshold, these layers can only further densify by the dry densification mechanism and not by water refreezing. Therefore, the overestimation of the ice slab thickness in the DPM profile is

compensated by the underestimation of its density, which leads to the good agreement with the observed FAC value. BK reproduces the presence of the ice slab at 1 m depth, but it underestimates its thickness and shows a large low-density section (Fig. 8a). Below the observed ice slab, the agreement with the average density is reasonable but the model underestimates density variability. Despite also underestimating the thickness of the ice slab, the R1M profile agrees better with the observed profile: it produces only two thin, low-density layers in the slab, and more high density peaks and ice layers below 7 m are in

better agreement with the observed density variability.

With respect to the 10 m temperature, the BK method gives results in reasonable agreement with the observations (-1.7 K), whereas the cold bias is more pronounced in R1M (-2.6 K). In contrast, DPM largely overestimates the 10 m temperature (+4.5 K), which stems from its overestimation of percolation and subsequent refreezing at great depths.



Changing the impermeability threshold for DPM (DPM wh02 ip780 and ip830) does not alter the general pattern of the modelled depth-density profile, but the density values of the ice slab become consistent with the impermeability threshold applied which affects the FAC accordingly (+15 % for ip780 and -9 % for ip830). Other factors further affect the FAC: runoff

rates slightly decrease with higher impermeability thresholds (Table 4) and the mass of the ice layers increases the overburden stress on the firn column below, increasing the densification rate. In addition, higher (lower) impermeability thresholds lead to warmer (colder) 10 m temperatures (-1.6 K for ip780 and +1.1 K for ip830), due to enhanced latent heat release. Compared to BK wh02 ip810, decreasing the impermeability threshold (BK wh02 ip780) leads to formation of ice layers in earlier years and closer to the surface and thus more runoff (+3 % of the water input over the entirety of the transient model run), which in

turn increases the FAC (+5 %) and decreases the 10 m temperature (-0.5 K). Increasing the threshold (BK wh02 ip830) has the opposite effect (-8 % for the FAC and +0.8 K for the 10 m temperature). If we instead allow for a greater water-holding capacity (BK whCL ip810), the partitioning between runoff and refreezing remains very similar (Table 4). However, the FAC and the 10 m temperature are changed (+3 % and -1.7 K). The lower temperature is due to latent heat release from refreezing being more concentrated in the surface layers (Fig. 6d). The formation of ice layers earlier in the year and at shallower depths

allows parts of the underlying firn to remain free of refreezing, which increases the FAC. Furthermore, colder temperatures cause a higher firn viscosity thus decreasing the densification rates. Since the R1M formulation both overestimates the FAC and underestimates the 10 m temperature, we test an increase in its impermeability threshold (R1M grLK ip830), allowing for deeper percolation. Both the decrease in FAC (-1 %) and increase in 10 m temperature (+0.1 K) are minor.

With the grA formulation in DPM (DPM grA ip810), water is more efficiently transferred vertically through the preferential flow domain, which causes an increase in the number of ice layers formed during the simulation (Fig. 6f), a slight decrease in FAC (-2 %) and a slight increase in the 10 m temperature (+0.1 K). The nearly-continuous ice slab, which extends to 17 m depth below the final winter accumulation, explains the weak sensitivity of the final FAC and 10 m temperature values of DPM to grain-size. In contrast, applying the grA formulation in R1M (R1M grA ip810) leads to a considerable increase in

FAC (+11 %) and a decrease in 10 m temperature (-0.7 K). This is due to higher water content during percolation events and, especially in the most recent years of our simulation, refreezing and ice-layer formation at shallower depths (Fig. 6e). This increases the runoff and isolates the deeper firn from meltwater percolation. As in the cases of DYE-2 and NASA-SE, the change in FAC due to different grain-size formulations in R1M is greater than the change due to switching from BK to R1M (Table 4).

The modelled depth-density profiles also differ according to the densification formulation used (Fig. 8b). We compare the different densification formulations using R1M grLK ip810, thus avoiding the effect of the strong temperature bias of DPM on the densification process. Densification in KM is sensitive to high firn temperatures, and it predicts the highest densities: it produces the highest density values in the ice slab range, the most ice layers below the ice slab and the lowest FAC value (-27



% compared to the CROCUS formulation). HL behaves in a similar way to CROCUS in the upper 5 m, apart from a much lower density interval in the 2-2.5 m depth range. In deeper firn, densities simulated using HL tend to lie between those simulated using KM and CROCUS, and its FAC difference with the CROCUS (-9 %) is less than that of KM. The DPM scheme simulates a depth-density profile of an ice slab over a 14 m range, which is in stark contrast with BK and R1M. Apart

from this, the choice of the densification formulation has a greater influence on the model than the choice of liquid water scheme and of any of their respective parameterisations presented here, in spite of the high water input at this site.

## 4.4 FA13

The FA13 site is representative of conditions in the southeast part of the GrIS; it has both high accumulation and high melt rates (mean 1980-2012 rates of 1.09 and 0.64 m w.e. yr$^{-1}$ respectively, Fig. 2). This favours the insulation of summer

percolating meltwater from winter atmospheric temperatures, typically leading to the formation of PFAs (Kuipers Munneke et al., 2014). Here, the initial conditions and the spin-up process cause the deep firn to be close to the melting point at the start of the transient run.

The warm firn, combined with the high water influx, allows liquid water to reach greater depths than at the other sites in all three flow schemes. Additionally, the firn – ice transition depth becomes important in the FA13 simulations. The observed

core shows that the 810 kg m$^{-3}$ density is reached and maintained from 24 m depth. The CROCUS densification scheme predicts that this density horizon occurs at 60 m depth. Since CROCUS has been developed for seasonal snow, the densification at high overburden stress is probably not well captured by the model (Stevens, 2018). Because of this, we base our simulations for FA13 on the HL densification model, which predicts this transition depth to be around 21 m.

The total refreezing rates are similar for the three flow schemes (Table 5). Since the deep firn is close to the melting point, the total refreezing amounts are essentially determined by the cold content provided in winter and the precise behaviour of the percolation has a minor impact. However, variability of refreezing with depth differs between schemes, which leads to differences in the 15 m FAC values (Table 5) and in the modelled depth-density profiles (Fig. 9a, b and c and Fig. 10a). FAC is consistently underestimated (-23 to -30 %) because firn density is overestimated above 10 m. R1M and DPM overestimate

density most strongly with FAC values 9 and 10 % smaller than BK respectively, and both schemes simulate the presence of a thick ice layer in the upper 10 m of the firn, which was not observed in the core. The BK model produces only a single thin ice layer in the 10 upper meters (0.2 m thick at 9 m depth), which is in good agreement with the observations (showing a single thin ice layer at 7.5 m depth). Below 10 m, the modelled densities are generally in better agreement and all the schemes produce several ice layers (Fig. 10a and Fig. 9a, b and c).

In the absence of any shallow ice layer throughout most of the simulation (Fig. 9a, b and c), meltwater is free to percolate through the winter accumulation layers and to deplete their cold content. The flow schemes have different abilities to store liquid water, which leads to small variations in runoff and refreezing rates. In BK, water is retained according to the water-



holding capacity (Fig. 11a) and refreezes during subsequent winters. In contrast, DPM allows percolation down to the firn – ice sheet transition where it ponds to form an aquifer (Fig. 11d and e). This leads to a significant reduction of runoff amounts during the aquifer build-up (-6 % of the water input over the entirety of the transient model run compared to BK) and the water remaining in the firn column is essentially constrained by the maximal amount of water we allow in the aquifer (1.65 m). In

5 theory, the same mechanism could be simulated by R1M but the percolating water is depleted before it reaches the bottom of the firn column (Fig. 11b). This is due to refreezing, to the lateral runoff parameterisation and to the presence of ice layers in the upper 10 m. No water persists through the winter seasons, which illustrates the model artefact that the effective saturation must be strictly positive for the stability of the RE (Sect. 3.2). Thus, the refreezing rates are slightly lower than in BK since no residual water is stored and later exposed to winter refreezing (Table 5).

The build-up of the aquifer starts very early (in the summer of 1981) when DPM is turned on in the transient run due to the low refreezing capacity of the deep firn. The depth of this aquifer is constrained by the impermeability threshold applied, which determines where the model places the firn ice transition. This depth is of 33 m in 1981 and 21 m in 2013, the decrease being caused by enhanced densification. The aquifer is fed only by preferential flow (Fig. 11e) since matrix flow cannot reach

the water table due to runoff, refreezing and the presence of ice layers in the firn column.
From 1994 and onwards the total simulated water content in summer is only regulated by the maximum allowed in the model (1.65 m). Since the water table is at a shallow depth towards the end of the simulation (7.5 m), the propagation from the surface of the cold winter temperatures can refreeze part of the saturated layers. This leads to the formation and progressive thickening of the shallow, thick ice layer. Also, the shallowness of the aquifer causes 23% of the porosity in the top 15 m to be filled with

20 liquid water and the 10 m temperature to be at the melting point.

The higher impermeability threshold in DPM grLK ip830 increases the depth of the calculated firn – ice transition, producing a deeper aquifer that extends between 12 and 29 m depth at the end of the simulation, compared to the 12-37 m depth range observed by Koenig et al. (2014). The increased depth leads to less refreezing in the shallowest layers of the aquifer and thus

a higher FAC value (+3 %) and to a 10 m temperature below the melting point.
The grain-size formulation following Arthern et al. (2010) (DPM grA ip810) reduces the ability of preferential flow to transport water down to the firn – ice transition but instead favours formation of discrete ice layers in the firn column (Fig. 8f). In this case the aquifer does not start to form until summer 1988, but the final aquifer structure (also between 7.5 and 21 m), the FAC value (-3 % for grA), and the partitioning between refreezing and runoff are similar to those simulated using grLK (Table 5).

In R1M, the sensitivity to grain-size is noticeable in the firn-structure evolution with differences in ice-layer formation between R1M grLK ip810 and R1M grA ip810 (Fig. 8b and e). The final FAC value (+4 % for grA) and the meltwater partitioning remain similar (Table 5), as for the case of DPM. This can be explained by the total refreezing's stronger dependence on the firn thermal structure than on the percolation pattern at this site.





Increasing the water-holding capacity in BK (BK whCL ip810) leads to a significantly lower FAC value (-12 %): more water refreezes in the near-surface layers, which reduces the runoff and enhances densification in the entire underlying firn column. Also, more water remains stored at depth throughout the different winter seasons (Fig. 11c), and some is still present at the end of the simulation (0.09 m) between 16 and 23 m depth. However, this small amount retained by the water-holding capacity

is much less than is stored in the saturated layers of the aquifer simulated in DPM.

We compare the three different densification models (CROCUS, HL, KM) using the R1M grLK ip810 flow scheme and these show important differences in the final modelled depth-density profiles (Fig. 10b). CROCUS agrees reasonably well with HL in the top 6 m but, as mentioned above, it has a strong low density bias at greater depths. Since CROCUS simulates lower

densification rates, its underestimation of the FAC value in the 15 upper meters (-21 %) is smaller than in HL  (-30 %), but it is clearly not representative of the density conditions below 15 m. KM predicts a firn column below the last winter's accumulation entirely at the ice density. The model thus identifies a firn – ice sheet transition at shallow depth (~2 m), which the water can reach before being depleted by the lateral runoff parameterisation and saturated layers can thus build up. This further amplifies the densification since the saturated layers at the transition depth are exposed to refreezing. Hence, in 2012,

runoff combined with refreezing exceeds the liquid water input (Table 5) since some layers wherein water had been stored in previous years reach the 810 kg m$^{-3}$ density, causing the stored water to be considered as runoff by the model. Whereas the FAC values are generally close for the different flow schemes and their parameterisations (maximal difference of 12 %), CROCUS and KM reach values 14 % higher and 34 % lower than HL respectively.

## 5 Discussion

The three liquid water schemes show consistent behaviour between sites. R1M generally predicts greater retention of water than the other schemes, which leads to more near-surface refreezing and thus more pore space and lower temperatures in deeper firn. In addition, the formation of ice layers close to the surface favours lateral runoff and thus contributes to lower densities and colder temperatures in the deep firn. As a result, when compared to observations R1M tends to reach higher FAC values and to underestimate 10 m temperatures. The BK formulation with the Coléou and Lesaffre (1998) parameterisation for

the water-holding capacity leads to the same effects, but they are amplified. The 10 m temperature biases are colder, suggesting that BK whCL does not allow for deep enough percolation. BK with the lower water-holding capacity (BK wh02) leads to a partitioning of the water input between refreezing and runoff similar to the more complex R1M at the four sites. As a result, the FAC values predicted by BK wh02 and R1M generally agree (maximum difference less than 10 %), as do the temperatures at 10 m depth (maximum difference less than 1 K). The FAC values and 10 m temperatures of R1M at the end of the model

runs always lie in the range of the ones obtained with different parameterisations of BK. This suggests that BK can produce results similar to R1M, provided it is parameterised appropriately. DPM shows a different behaviour: it effectively brings water to greater depths and depletes the deep-firn pore space and cold content. Even in the presence of shallow ice layers



hindering matrix flow, the preferential flow implementation still ensures efficient vertical water transport, and runoff amounts remain low. This suggests that transfer mechanisms to the preferential flow domain implemented in DPM are more effective in draining ponding water than the lateral runoff parameterisation. Due to large FAC underestimation and 10 m temperature overestimation, the data-model mismatch of DPM is significantly greater than that of R1M and BK. However, DPM is better

at producing density variability in depth, which is underestimated in all schemes at all sites. Also, in contrast to the two other schemes, DPM can form ice layers even in summers of average melt, and it is able to simulate the persistence of deep saturated firn layers at the FA13 site. It is important to bear in mind that we only use the flow scheme of SNOWPACK in DPM; the results produced by the full SNOWPACK model would be different because it has its own formulations for snow mechanical and thermal properties. In particular, DPM relies heavily on the grain-size, and it would thus benefit from better representations

of the firn's structural properties. Moreover, the primary purpose of the DPM implementation in SNOWPACK is to reproduce the occurrence of ice layers in a seasonal alpine snow pack (Wever et al., 2016), whereas in this study we evaluate its ability to simulate representative firn depth-density profiles over the course of numerous decades.

The lack of density variability in the modelled profiles cannot only be attributed to inaccuracies in the percolation-refreezing

process. This is demonstrated in the example of NASA-SE: the layers of the density peak observed around 1 m depth (Fig. 4a and b) were deposited during the final winter of the simulation (2015-2016). As such, these have only been influenced by the percolation and refreezing of negligible amounts of liquid water. The consistent underestimation of density variability across all schemes indicates that one or several other factors that are not or poorly represented by firn models likely play a crucial role in firn evolution. These factors may include horizontal water flow, variable density of fresh snow, effects of firn

microstructure on densification, impurity content, wind packing and short-term weather fluctuations. Moreover, the validity of the firn model relies on the accuracy of the climatic forcing.

At KAN-U, despite imperfect agreement with the observed density profile (Fig. 7a), DPM predicts the 15 m FAC accurately (Table 4). This could suggest that it predicts the correct amount of refreezing at this site integrated over the top 15 m. On the

other hand, this model-data agreement could result from numerous errors in the model compensating for each other. DPM strongly overestimates temperature at 10 m depth, suggesting that this refreezing is occurring too deep in the firn column. It likely overestimates the percolation whereas in reality, water may pond for longer in soaked firn close to the surface (Pfeffer and Humphrey, 1996). The subsequent refreezing allows for more of the released latent heat to be dissipated towards the atmosphere. It is also possible that DPM overestimates the total refreezing and that enhanced densification causes part of the

FAC depletion. The existing firn-densification formulations are likely not suited for representing densification in conditions of high water contents and high refreezing rates. Our study indicates that firn-densification models could be improved by accounting for the latent heat source as well as the effects of liquid water and of refreezing cycles on firn viscosity and densification rates. For example, the KM and HL densification equations were established for dry firn (Herron and Langway, 1980; Kuipers Munneke et al., 2015a). In the CROCUS scheme, firn viscosity is adjusted according to the water content, but



our results show that the modification in the parameterisation is insufficient to reproduce the observed densities at KAN-U. At all sites, interchanging the HL, KM and CROCUS formulations for firn densification generally leads to more variability in the results than using different water flow schemes. A simple example of the densification schemes in HL and KM not representing reality becomes apparent when applying the percolation-refreezing schemes: in reality, densification is dependent on the

overburden stress, but these models use accumulation rate as a proxy for stress. Consequently, in these models the redistribution of mass due to runoff and percolation does not affect the densification rates, despite the effect it has on the firn column mass. The absence of a preferential flow scheme is often presented as a possible explanation for firn-density overestimation close to the surface (e.g. Gascon et al., 2014; Kuipers Munneke et al., 2014; Steger et al., 2017). However, our results suggest that simply adding a one-dimensional preferential flow scheme, although physically detailed, to firn-densification models does not

solve this issue. The water that is transported quickly from the shallow layers must flow back into the matrix domain at some point. If this occurs in the shallow layers, the density-overestimation issue remains; if this occurs in deep layers it can lead to unrealistic temperature signatures. The representation of preferential flow physics requires improvements and there are several other possible factors for densification errors at such high melt sites, including exaggerated sensitivity of the model to temperature.

Both DPM and R1M exhibit significant sensitivity to the choice of the grain-size formulation (grLK or grA). Modifying this formulation in R1M affects the model results more than changing to the use of BK at all sites apart from FA13 where the magnitudes of change are comparable. This highlights another significant difficulty for percolation schemes: the dependence of water flow on the firn's structural properties. Field evidence demonstrates the crucial role of structural transitions, even at

the scale of centimeters, on the behaviour of water flow in firn (Marsh and Woo, 1984; Pfeffer and Humphrey, 1996, 1998; Williams et al., 2010). With respect to this, the advanced flow schemes applied in this study have some limitations. Firstly, the structural properties of grains in the firn layer are poorly constrained by observations. Secondly, the parameterisations linking the structural and hydraulic properties on which R1M and DPM rely were derived from a limited number of laboratory experiments. These are typically performed at a very small scale (e.g. shallow snow columns with diameter of 5 cm in the

experiments of Yamaguchi et al. (2012)), and are mostly based on homogeneous snow in terms of grain-size and temperature (Yamaguchi et al. 2012; Katsushima et al., 2013). The much larger scale of the GrIS firn layer, the spatial and temporal heterogeneity of its structural properties, and its climatic and glaciological settings render the validity of these idealised parameterisations questionable. Finally, the density dependence of the parameters makes them sensitive to errors in the densification process. Thus, a better knowledge of firn structural properties would only be profitable to flow schemes if we

have a clear understanding of the link between snow structure and its hydraulic properties and vice versa.

Another major limitation of the implementation of physically detailed liquid water flow schemes in one-dimensional firn models is the fact that water flow is in reality three dimensional. Water can flow horizontally on top of buried ice lenses or on thin, near-surface ice crusts caused by daily refreezing (Marsh and Woo, 1984; Pfeffer and Humphrey, 1996). Even at depths



greater than 10 m, large masses of liquid water can persist through the winter and move laterally over considerable distances (Humphrey et al., 2012). In one-dimensional models, the key to solving this issue is to accurately partition between vertical percolation and lateral flow; this likely requires a better approach than the lateral runoff parameterisation we implement here. As an example, at FA13 all three water-transport schemes overestimate the density in the 10 upper meters except in the last

winter's accumulated layers (0 – 2 m depth), where there is a good agreement with observations (Fig. 10a). This suggests a consistent overestimation of the summer meltwater refreezing and underestimation of lateral runoff. Also, the need to use a limit for the PFA water content demonstrates that some processes not represented in the model must regulate its water volume; these are likely lateral movement driven by hydraulic pressure gradients and connections with englacial and subglacial hydrological systems. A one-dimensional preferential flow scheme aims to correctly partition the water input between matrix

flow and fast preferential flow; there are several other difficulties with this approach. These include accurately determining how deep the water can be transported by preferential flow, how much water refreezes and how much is stored as liquid water, and the amount of lateral flow at different depths in the firn column. Similarly, the same considerations apply to liquid water flowing in from upstream grid cells.

Our results also suggest that more observations of firn-temperature variability in time and depth would likely be useful for the evaluation of existing flow schemes and the development of new ones. Modelled temperature profiles show both the depth and volume of refreezing due to the release of latent heat. Moreover, deep meltwater refreezing causes marked and long-lasting temperature increases due to insulation from the overlying firn, and temperature measurements in depth can be powerful indicators of the occurrence of deep percolation and refreezing events. On the other hand, comparisons between modelled and

observed density profiles are strongly affected by the choice and accuracy of the densification formulation, the variability of surface density, several other factors influencing model outputs mentioned above and possible uncertainties in field measurements. However, the modelled temperature profile also depends on the accuracy of the climatic forcing, of the heat-transport scheme and of the thermal-conductivity parameterisation. The latter is a function of density, thus erroneous depth-density profiles inevitably lead to an inaccurate heat-transport process. As an example of the influence of various sources of

errors, the warm 10 m temperature bias of all the schemes at NASA-SE (Table 3) is unlikely to be only due to the percolation-refreezing process.

## 6 Conclusion

We implemented three liquid water schemes of different levels of physical complexity in a firn model using a fine vertical resolution: a bucket scheme, Richards Equation in a matrix flow scheme, and Richards Equation in a preferential flow scheme.

To our knowledge, this is the first study to apply the Richards Equation as well as a preferential flow scheme in firn-densification simulations on the GrIS.




Our three liquid water flow schemes predict significantly different vertical patterns of refreezing and consequently modelled densities, firn air content values and 10 m temperatures. The preferential flow scheme effectively evacuates meltwater from the surface layers and leads to underestimation of firn air content and overestimation of 10 m temperatures. Compared to the preferential flow scheme, the single-domain Richards Equation scheme generally showed biases of the opposite signs and of much lower magnitudes, suggesting it slightly underestimates percolation depths. The simpler bucket scheme predicted refreezing rates, firn air contents and 10 m temperatures similar to those obtained by the single-domain Richards Equation; by adjusting its water-holding capacity and impermeable density parameters, it could produce the same results. Using the Coléou and Lesaffre (1998) parameterisation for the water-holding capacity in the bucket scheme led to underestimation of percolation depths. The bucket scheme with lower water-holding capacity and the single-domain Richards Equation scheme predicted firn air contents and 10 m temperatures in closest agreement with observations. However, the preferential flow scheme was found to perform better than the simpler flow schemes in reproducing the density variability with depth and the water-saturated conditions at the bottom of the firn column at a site of a perennial firn aquifer.

We identified the multidimensionality of liquid water flow as the prominent challenge for water percolation schemes. Because firn models are currently one dimensional, an accurate partitioning between horizontal and vertical flow is likely to be at least as difficult and as important as the separation between slow matrix and rapid preferential flow. Other difficulties related to water-flow representation include the uncertainties in firn hydraulic properties and in firn micro- and macro-structure on the GrIS. This is further demonstrated by our results showing the sensitivity of the Richards Equation-based schemes on the grain-size formulation. However, the absence of any large-scale field observations of water flow in firn makes it difficult to constrain its implementation and to validate model behaviour. By using flow schemes developed for snow models, the goal of this study was to identify limitations in implementing such schemes in firn and research needed to improve liquid water schemes. Whilst we did apply some modifications to account for the differences between snow and firn, we suggest that more modifications are likely required since the spatial scales and the structural characteristics of seasonal snowpacks and firn are different.

There are a number of effects that influence firn density, which hamper the validation of a particular flow scheme based on observed depth-density profiles. As an example, the density variability in depth was largely underestimated regardless of the flow scheme. This suggests that there are uncaptured complexities in the percolation and refreezing mechanisms that need to be incorporated into models and also that firn-model development must focus on including complex processes currently poorly or not represented, such as surface-density variability and firn structural effects on densification. A comprehensive exploration of the various firn models and their parameter spaces could help identify priorities for further model developments based on minimising data-model mismatch and overall uncertainty. In line with this, we showed that output from three common firn-densification models shows greater variability than the output from single densification model using the different flow schemes. In order to capture the multiple impacts of liquid water on firn densification, future models require an improved liquid water flow scheme, accurate boundary conditions, and formulations developed explicitly to simulate densification of wet firn.



*Author contributions.* VV and AL conceived this study. VV performed the development of the water flow schemes, performed the model experiments and led writing of the manuscript. AL supervised the work. MS contributed to development of the water flow schemes and model experiments. MMF provided the firn core data of the FirnCover Project. BN and MRB provided the RACMO2.3 forcing data. All authors provided comments and suggested edits to the manuscript.

*Acknowledgements.* We thank Laura Koenig for making available the FA13 firn core data in the SUMup dataset. We thank Horst Machguth for making available the KAN-U firn core data. We thank the Geological Survey of Denmark and Greenland for making temperature measurements at KAN-U available in the PROMICE dataset. Keith Beven is acknowledged for insight into the subject of hydrology. Nander Wever is acknowledged for support in the understanding of the SNOWPACK model.

Part of the funding for this research was provided by the Centre for Polar Observation and Modelling.

*Competing interests.* The authors declare that they have no conflict of interest.

## Appendix A: Model Implementation

The model uses a finite-volume scheme with each layer being an independent volume. We use the general mixed-form Picard
iteration scheme to solve the RE, as it has been demonstrated that the mixed form of RE can be efficiently used in finite-difference schemes because of its accuracy and its robustness with respect to mass conservation (Celia et al., 1990). The Picard scheme discretises the model using central finite differences for the space derivative and a backward Euler method for the time derivative. The iterative process calculates the value of the pressure head at each iteration and then adjusts the liquid water content according to the water retention curve, Eq. (12). Hydraulic parameters are updated and iterations are repeated until
convergence of the solution is achieved. Boundary conditions are the rate of meltwater input at the surface and a no-flow condition at the bottom. Solving the RE in the firn column presents numerical challenges. We adopt an implementation strategy based on the works of Wever et al. (2014) and D'Amboise et al. (2017) who implemented the RE in the snow models SNOWPACK and CROCUS respectively. Here, we give more details about this methodology.

### A1 Convergence criteria

For the solution reached by the Picard iteration scheme to be considered convergent, it must fulfil different criteria. The convergence criteria between two successive iterations are defined for the head pressure ($\varepsilon_h$) and liquid water content ($\varepsilon_\theta$) values as well as for the mass balance error ($\varepsilon_{MB}$) of individual layers. These three criteria are fixed to $10^{-3}$ m, $10^{-5}$ and $10^{-8}$ m respectively, following Huang et al. (1996) and Wever et al. (2014). For each layer, we select $\varepsilon_\theta$ or $\varepsilon_h$ according to the effective saturation. Huang et al. (1996) showed that using $\varepsilon_\theta$ allows faster convergence. However, it cannot be used in very saturated
layers and thus we apply $\varepsilon_h$ for layers where effective saturation exceeds 0.99, in accordance with Wever et al. (2014). The mass-balance criterion is always applied to every layer, regardless of the saturation.



## A2 Hydraulic conductivity calculation

As we use a central finite difference approach to compute RE, the fluxes are assumed to occur on the interface between adjacent layers. Incoming and outgoing fluxes are computed and this requires the hydraulic conductivity value to be calculated at the top and bottom of every layer $i$ and not at the centre. We use the upstream-weighting technique (Forsyth et al., 1995):

$$5 \quad K_{i+\frac{1}{2}} = \begin{cases} K_i, & if \ \frac{\Delta h}{\Delta z} - 1 \leq 0 \\ K_{i+1}, & if \ \frac{\Delta h}{\Delta z} - 1 > 0 \end{cases} \quad \text{(A1)}$$

The advantage of this formulation over a simple arithmetic mean is that it does not lead to oscillatory solutions, regardless of the mesh size (Forsyth et al., 1995; Szymkiewicz, 2009).

## A3 Dry layers

For numerical stability, a snow layer cannot be completely dry (i.e. $\theta = 0$). Therefore, two cases must be considered: dry layers and refreezing layers. At the start of the flow routine, all layers are initialised with a very low $\theta$ value, $\theta_{dry}$. The value must be sufficiently low to avoid influencing the refreezing process but sufficiently high to lead to a convergent solution (D'Amboise et al., 2017). In this study, the $\theta_{dry}$ value is fixed at $10^{-6}$ as this is a tenth of the $\varepsilon_\theta$ criterion. This corresponds to a 1 m thick snow layer holding 1 µm of liquid water. When the flow routine is called in the firn model, the water content of every dry layer is thus synthetically raised to $\theta_{dry}$, which corresponds to a pre-wetting. The porosity of ice layers that are at high densities (>900 kg m$^{-3}$) is thus adjusted in order to raise their water content to $\theta_{dry}$ in both domains.

Similarly, there is a risk for $\theta$ reaching too-low values when refreezing occurs. Therefore, refreezing is allowed only if the $\theta$ value is above 0.01% (Wever et al., 2014). This value is above $\theta_{dry}$ to avoid refreezing and corresponding latent heat release of the very low amounts of water resulting from the fluxes between layers that are initialised at $\theta_{dry}$. Only at the last time step of the flow routine is the refreezing process allowed to decrease the volumetric water content until $\theta_{dry}$. After that, the pre-wetting amounts of liquid water are subtracted at the end of the flow routine to maintain the mass conservation property of the firn model. At the end of the flow routine, if all the layers have a water content below $\varepsilon_\theta$, we consider the firn column to be completely dry again so that the flow routine does not have to be called until the next melt event and computational time is largely saved.

## A4 Dynamical time step adjustment

The numerical solving of RE uses a dynamically adjusted time step. Certain situations, such as the arrival of the wetting front at a stratigraphic transition, require a very small time step whereas larger time steps can be used in other cases without affecting numerical stability. Thus, the time step is adjusted according to the number of iterations, $n_{it}$, required to achieve convergence of the solution at the previous time step: decreased for a large number of iterations and increased for few iterations. Also, as in Wever et al. (2014) and D'Amboise et al. (2017), a back step case is used: the calculation is stopped and the time step automatically decreased if the solution fails to converge in 15 iterations or if warning signs of instability appear (positive





pressure head values, effective saturation exceeding 1 or differences in successive pressure head values exceeding $10^3$ m). The time step is bounded between $10^{-20}$ s and 900 s. The procedure can be summarised as follows:

$$\Delta t_{RE}^t = \begin{cases} 1.25 \, \Delta t_{RE}^{t-1}, & if \; n_{it} \leq 5 \\ \Delta t_{RE}^{t-1} & , \; if \; 5 < n_{it} < 10 \\ 0.5 \, \Delta t_{RE}^{t-1} & , \; if \; 10 \leq n_{it} \leq 15 \\ back \; step & , \; if \; n_{it} > 15 \end{cases} \tag{A2}$$

**A5 Saturated layers and aquifer treatment**

5   If water reaches an impermeable ice layer, the layer above progressively becomes saturated. This means that its hydraulic conductivity progressively increases. As a consequence, the incoming flow becomes very large whereas the outgoing flow is forced to be zero. To deal with this issue, the layer has to be set impermeable once close to saturation and this process must go on for layers above when these reach saturation in turn. When an aquifer is present at the bottom of the domain, the amount of water is held in memory at the start of the flow routine and the end of the domain is set as the top of the aquifer. All the percolating water reaching the end of the domain is added to the aquifer amount and at the end of the routine, this total amount is redistributed in the bottom layers.

**A6 Partial RE solving**

In order to save computational time, the RE is not necessarily solved for the entire domain. If a significant part of the lowest layers is dry, we do not proceed to the calculations for this lower dry part. Starting from the surface, we look for the lowest layer where the water content is at least $\varepsilon_\theta + 0.01$ % (above the minimum water content after refreezing). Then we take as lower limit for the RE calculation the layer situated 50 cm below this lowest wet layer. This is recalculated at every time step of the RE solving, making the 50 cm addition largely sufficient to capture the wetting of the dry lower part. If the lowest wet layer is less than 50 cm above the end of the domain, then the RE is calculated on the entire domain. This is applied in both the matrix and the preferential flow domains.

**A7 Refreezing in the preferential flow domain**

Contrarily to the SNOWPACK model, there is a particular circumstance for which we apply refreezing directly in the preferential flow domain: if a cold front (subfreezing temperatures) propagates from the surface into a wet firn column, all the water present in the matrix flow domain will progressively be refrozen, starting from the surface layer. It would be unrealistic to keep liquid water present in the preferential flow domain of layers that are above this cold front. Thus, if starting from the surface, the entire firn column until a particular layer that holds some liquid water in the preferential flow domain is at subfreezing temperatures, this liquid water is refrozen. In such cases, the firn column is dry in both domains until the depth delimited by the cold front. Simulations without this refreezing implementation reached very similar results but required more computational time.




## A8 Merging process

The CFM usually considers every accumulation event as a new layer. However, as we use three hourly accumulation forcing, the firn layer could consist of a high number of extremely fine layers. Because the calculation time for the RE is very dependent on the number of distinct layers in the firn column, we chose to merge any layer thinner than 2 cm with the underlying layer. If this was applied to the surface layer, every accumulation event of less than 2 cm snow would be immediately merged with the previous surface layer. In the case where a high number of successive snowfall events would be below the 2 cm threshold, these would all be merged within the same layer, possibly becoming very thick. To avoid this, the newly added snow layer is merged with the previous surface layer only if the latter is below the 2 cm threshold. However, newly-added layers that are less than 0.01 mm thick are always merged with the layer below. It is important to keep a high vertical resolution when simulating the percolation process with the RE, as this flow equation is highly sensitive to structural heterogeneities in the firn. If the merging process is too lenient, this leads to the smoothing of heterogeneities such as sharp grain-size or density transitions. Moreover, using a coarse resolution would lead to only an approximation of the water percolation because water content is always homogeneous in a single layer. Thus, as soon as water percolates at the top of a given layer, it is distributed in the entire layer.

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



| Bucket Model (BM) | | | | | Single-domain RE (R1M) | | | | | Dual-Permeability RE (DPM) | | | | |
|---|---|---|---|---|---|---|---|---|---|---|---|---|---|---|
| Impermeability threshold (ip) | | | Water-holding capacity (wh) | | Impermeability threshold (ip) | | | Grain-size formulation (gr) | | Impermeability threshold (ip) | | | Grain-size formulation (gr) | |
| 780 | 810 | 830 | 0.02 | CL | 780 | 810 | 830 | LK | A | 780 | 810 | 830 | LK | A |

**Table 1. Summary of the sensitivity tests**

| | Refreezing / Inflow (1980-2011) | Refreezing / Inflow (2012-2016) | Runoff / Inflow (1980-2011) | Runoff / Inflow (2012-2016) | Top 15 m FAC [m] (anomaly vs observations) | T 10m [K] (anomaly vs observations [K]) |
|---|---|---|---|---|---|---|
| BK (wh02 ip810) | 0.96 | 0.67 | 0.01 | 0.31 | 5.01 (-4 %) | 260.88 (+0.21) |
| R1M (grLK ip810) | 0.91 | 0.63 | 0.05 | 0.35 | 4.99 (-4 %) | 260.27 (-0.40) |
| DPM (grLK ip810) | 0.95 | 0.95 | 0.02 | 0.03 | 4.38 (-16%) | 263.39 (+2.72) |
| Observations | / | / | / | / | 5.21 | 260.67 |
| DPM (grLK ip780) | 0.95 | 0.96 | 0.02 | 0.02 | 4.39 (-16%) | 263.45 (+2.78) |
| BK (wh2 ip780) | 0.96 | 0.60 | 0.02 | 0.38 | 5.15 (-1 %) | 260.68 (+0.01) |
| BK (whCL ip810) | 0.92 | 0.62 | 0.05 | 0.36 | 5.21 (+0 %) | 259.40 (-1.27) |
| BK (wh02 ip830) | 0.97 | 0.68 | 0.00 | 0.31 | 4.96 (-5 %) | 260.98 (+0.31) |
| R1M (grA ip810) | 0.83 | 0.59 | 0.14 | 0.38 | 5.19 (-0 %) | 259.64 (-1.03) |
| DPM (grA ip810) | 0.93 | 0.93 | 0.04 | 0.05 | 4.40 (- 16%) | 263.08 (+2.41) |
| HL DPM (grLK ip810) | 0.95 | 0.96 | 0.02 | 0.02 | 4.16 (-20 %) | 263.78 (+3.11) |
| KM DPM (grLK ip810) | 0.95 | 0.95 | 0.02 | 0.02 | 3.35 (-36%) | 262.74 (+2.07) |
| HL R1M (grLK ip810) | 0.90 | 0.72 | 0.07 | 0.26 | 4.65 (-11 %) | 260.41 (-0.26) |
| KM R1M (grLK ip810) | 0.91 | 0.75 | 0.06 | 0.23 | 4.00 (-23 %) | 260.26 (-0.41) |

**Table 2. Model outputs at DYE-2 site. / indicates no data.**

| | Refreezing / Inflow (1980-2011) | Refreezing / Inflow (2012-2015) | Runoff / Inflow (1980-2011) | Runoff / Inflow (2012-2015) | Top 15 m FAC [m] (anomaly vs observations) | T 10m [K] (anomaly vs observations [K]) |
|---|---|---|---|---|---|---|
| BK (wh02 ip810) | 0.97 | 0.97 | 0.00 | 0.00 | 6.78 (-3 %) | 257.91 (+1.94) |
| R1M (grLK ip810) | 0.94 | 0.89 | 0.02 | 0.08 | 6.78 (-3 %) | 257.39 (+1.42) |
| DPM (grLK ip810) | 0.95 | 0.94 | 0.01 | 0.03 | 6.77 (-3 %) | 258.18 (+2.21) |
| Observations | / | / | / | / | 6.98 | 255.97 |
| BK (whCL ip810) | 0.95 | 0.94 | 0.00 | 0.02 | 6.81 (-2 %) | 256.97 (+1.00) |
| R1M (grA ip810) | 0.92 | 0.83 | 0.04 | 0.13 | 6.83 (-2 %) | 256.99 (+1.02) |
| DPM (grA ip810) | 0.95 | 0.92 | 0.01 | 0.04 | 6.78 (-3 %) | 258.11 (+2.14) |
| HL R1M (grLK ip810) | 0.93 | 0.86 | 0.04 | 0.11 | 8.13 (+17 %) | 258.19 (+1.22) |
| KM R1M (grLK ip810) | 0.93 | 0.86 | 0.03 | 0.10 | 7.53 (+8 %) | 257.74 (+1.77) |

**Table 3. Model outputs at NASA-SE site. / indicates no data.**

| | Refreezing / Inflow (1980-2011) | Refreezing / Inflow (2012) | Runoff / Inflow (1980-2011) | Runoff / Inflow (2012) | Top 15 m FAC [m] (anomaly vs observations) | T 10m [K] (anomaly vs observations [K]) |
|---|---|---|---|---|---|---|
| BK (wh02 ip810) | 0.81 | 0.18 | 0.17 | 0.81 | 3.92 (+59%) | 263.93 (-1.73) |
| R1M (grLK ip810) | 0.74 | 0.20 | 0.23 | 0.79 | 3.69 (+50 %) | 263.09 (-2.57) |
| DPM (grLK ip810) | 0.91 | 0.77 | 0.07 | 0.23 | 2.40 (-2 %) | 270.18 (+4.52) |
| Observations | / | / | / | / | 2.46 | 265.66 |
| DPM (grLK ip780) | 0.91 | 0.75 | 0.07 | 0.25 | 2.77 (+13 %) | 268.63 (+2.97) |
| DPM (grLK ip 830) | 0.90 | 0.82 | 0.07 | 0.18 | 2.18 (-11%) | 271.31 (+5.65) |
| BK (wh02 ip780) | 0.78 | 0.19 | 0.21 | 0.81 | 4.11 (+67 %) | 263.46 (-2.20) |
| BK (whCL ip810) | 0.78 | 0.22 | 0.19 | 0.78 | 4.05 (+65 %) | 262.23 (-3.43) |
| BK (wh02 ip830) | 0.83 | 0.21 | 0.15 | 0.79 | 3.61 (+47 %) | 264.69 (-0.97) |
| R1M (grLK ip830) | 0.76 | 0.20 | 0.22 | 0.79 | 3.64 (+48 %) | 263.21 (-2.45) |
| R1M (grA ip810) | 0.66 | 0.26 | 0.31 | 0.73 | 4.08 (+66 %) | 262.41 (-3.25) |
| DPM (grA ip810) | 0.87 | 0.69 | 0.10 | 0.30 | 2.36 (-4 %) | 270.28 (+4.62) |
| HL R1M (grLK ip810) | 0.74 | 0.18 | 0.23 | 0.82 | 3.36 (+37 %) | 263.34 (-2.32) |
| KM R1M (grLK ip810) | 0.74 | 0.20 | 0.23 | 0.80 | 2.70 (+10%) | 262.82 (-2.84) |

**Table 4. Model outputs at KAN-U site. / indicates no data.**

| | Refreezing / Inflow (1980-2011) | Refreezing / Inflow (2012) | Runoff / Inflow (1980-2011) | Runoff / Inflow (2012) | Top 15 m FAC [m] (anomaly vs observations) | T 10m [K] (anomaly vs observations [K]) | Remaining water [m] |
|---|---|---|---|---|---|---|---|
| BK (HL wh02 ip810) | 0.55 | 0.28 | 0.45 | 0.73 | 3.82 (-23 %) | 271.75 (+0.10) | 0 |
| R1M (HL grLK ip810) | 0.50 | 0.28 | 0.49 | 0.71 | 3.47 (-30 %) | 270.94 (-0.71) | 0 |
| DPM (HL grLK ip810) | 0.51 | 0.32 | 0.38 | 0.70 | 3.45 (-30 %) of which 0.81 m of water | 273.15 (+1.5) | 1.53 |
| Observations | / | / | / | / | 4.96 | 271.65 | 1.65 |
| BK (HL whCL ip810) | 0.60 | 0.23 | 0.38 | 0.81 | 3.38 (-32 %) | 270.99 (-0.66) | 0.09 |
| R1M (HL grA ip810) | 0.46 | 0.30 | 0.52 | 0.69 | 3.60 (-27 %) | 269.77 (-1.88) | 0 |
| DPM (HL grA ip810) | 0.51 | 0.36 | 0.39 | 0.70 | 3.36 (-32 %) of which 0.83 m of water | 273.15 (+1.5) | 1.48 |
| DPM (HL grLK ip830) | 0.51 | 0.29 | 0.39 | 0.70 | 3.57 (-28 %) of which 0.38 m of water | 272.13 (+0.48) | 1.64 |
| CROCUS R1M (grLK ip810) | 0.49 | 0.31 | 0.49 | 0.68 | 3.94 (-21%) | 271.46 (-0.19) | 0 |
| KM R1M (grLK ip810) | 0.51 | 0.37 | 0.45 | 0.99 | 2.29 (-54 %) | 270.90 (-0.75) | 0 |

**Table 5. Model outputs at FA13 site. / indicates no data.**



| Variable/Parameter | Symbol | Value [unit] |
|---|---|---|
| Density | $\rho$ | [kg m$^{-3}$] |
| Ice density | $\rho_i$ | 917 [kg m$^{-3}$] |
| Water density | $\rho_w$ | 1000 [kg m$^{-3}$] |
| Temperature | $T$ | [K] |
| Mean annual surface temperature | $T_{av}$ | [K] |
| Mean annual accumulation rate | $\dot{b}$ | [m s$^{-1}$] |
| Gas constant | $R$ | 8.314 [J mol$^{-1}$ K$^{-1}$] |
| Gravitational acceleration | $g$ | 9.81 [m s$^{-2}$] |
| Overburden pressure | $\sigma$ | [kg m$^{-1}$ s$^{-2}$] |
| Snow viscosity | $\eta$ | [kg m$^{-1}$ s$^{-1}$] |
| | $\eta_0$ | 7.62237 [kg s$^{-1}$ m$^{-1}$] |
| | $a_n$ | 0.1 [K$^{-1}$] |
| | $b_n$ | 0.023 [m$^3$ kg$^{-1}$] |
| Firn viscosity parameters | $c_n$ | 358 [kg m$^{-3}$] |
| | $f_1$ | 4 [/] |
| | $f_2$ | [/] |
| Firn thermal conductivity | $k_s$ | [W m$^{-1}$ K$^{-1}$] |
| Pressure head | $h$ | [m] |
| Hydraulic conductivity | $K(\theta)$ | [m s$^{-1}$] |
| Hydraulic conductivity at saturation | $Ksat$ | [m s$^{-1}$] |
| Grain radius | $r$ | [m] |
| Grain radius at surface | $r_0$ | [m] |
| Grain growth activation energy | $E_g$ | 42.4 10$^3$ [J mol$^{-1}$] |
| Grain growth rate constant | $k_g$ | 1.3 10$^{-7}$ [m$^2$ s$^{-1}$] |
| | $b_0$ | 0.781 [/] |
| Initial grain-size parameters | $b_1$ | 0.0085 [/] |
| | $b_2$ | -0.279 [/] |
| Dynamic viscosity of liquid water at 273.15 K | $\mu$ | 0.001792 [kg m$^{-1}$ s$^{-1}$] |
| Volumetric water content | $\theta$ | [/] |
| Water-holding capacity | $\theta_h$ | [/] |
| Mass proportion corresponding to water-holding capacity | $W_w$ | [/] |
| Residual water content | $\theta_r$ | [/] |
| Porosity | $P$ | [/] |
| Fraction of the pore space allocated to preferential flow | $F$ | 0.02 [/] |
| Saturated water content | $\theta_{sat}$ | [/] |
| Effective saturation | $Se$ | [/] |
| van Genuchten parameters | $\alpha, n, m$ | [/] |
| Water entry suction | $h_{we}$ | [m] |
| Heat flow | $Q$ | [J m$^{-2}$ s$^{-1}$] |
| Specific latent heat of fusion | $L_f$ | [J kg$^{-1}$] |



| | | |
|---|---|---|
| Concentration of preferential flowpaths | $N$ | [m$^{-2}$] |
| Preferential flow saturation threshold | $\Theta$ | 0.1 [/] |
| Lateral runoff | $Ru$ | [m] |
| Water in excess of the residual water content | $L_{excess}$ | [m] |
| Characteristic runoff time | $\tau_{Ru}$ | [s] |
| Surface slope | $S$ | [/] |
| | $c_1$ | 1.296 10$^5$ [s] |
| Runoff parameters | $c_2$ | 2.16 10$^6$ [s] |
| | $c_3$ | 140 [/] |
| Mass liquid water content | $\theta_{weight,\%}$ | [%] |

**Table A1. Variables and parameters notation**

| | |
|---|---|
| Liquid water scheme [Abbreviation] | Bucket model [BK]; Single-domain Richards Equation [R1M]; Dual-permeability Richards Equation [DPM] |
| Compaction scheme [Abbreviation] | CROCUS [CR]; Herron and Langway (1980) [HL]; Kuipers Munneke et al. (2015) [KM] |
| Impermeability threshold [Abbreviation] | 780 kg m$^{-3}$ [ip780]; 810 kg m$^{-3}$ [ip810]; 830 kg m$^{-3}$ [ip830] |
| Water-holding capacity [Abbreviation] | Constant at 2 % [wh02]; Coléou and Lesaffre (1998) [whCL] |
| Grain-size formulation [Abbreviation] | Linow et al. (2012) at surface and Katsushima et al. (2009) growth [grLK]; Constant at surface and Arthern et al. (2010) growth [grA] |

**Table A2. Options for simulation experiments and their respective abbreviations**

| | Liquid water scheme | Compaction scheme | Impermeability threshold | Water-holding capacity | Grain-size formulation |
|---|---|---|---|---|---|
| BK (wh02 ip810) | BK | CROCUS | 810 | 0.02 | / |
| R1M (grLK ip810) | R1M | CROCUS | 810 | / | LK |
| DPM (grLK ip810) | DPM | CROCUS | 810 | / | LK |
| DPM (grLK ip780) | DPM | CROCUS | 780 | / | LK |
| DPM (grLK ip830) | DPM | CROCUS | 830 | / | LK |
| BK (wh02 ip780) | BK | CROCUS | 780 | 0.02 | / |
| BK (whCL ip810) | BK | CROCUS | 810 | CL | / |
| BK (wh02 ip830) | BK | CROCUS | 830 | 0.02 | / |
| R1M (grLK ip830) | R1M | CROCUS | 830 | / | LK |
| R1M (grA ip810) | R1M | CROCUS | 810 | / | A |
| DPM (grA ip810) | DPM | CROCUS | 810 | / | A |
| HL DPM (grLK ip810) | DPM | HL | 810 | / | LK |
| KM DPM (grLK ip810) | DPM | KM | 810 | / | LK |
| HL R1M (grLK ip810) | R1M | HL | 810 | / | LK |
| KM R1M (grLK ip810) | R1M | KM | 810 | / | LK |

**Table A3. Details of the acronyms of the simulation experiments presented**





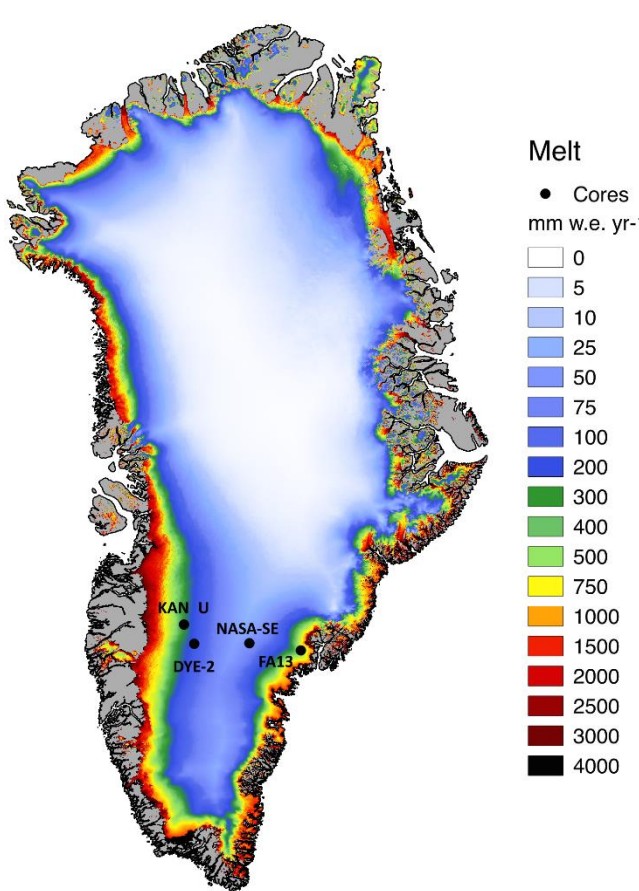

Figure 1. Study sites locations and mean annual melt rates (1958-2017) from RACMO2.3p2



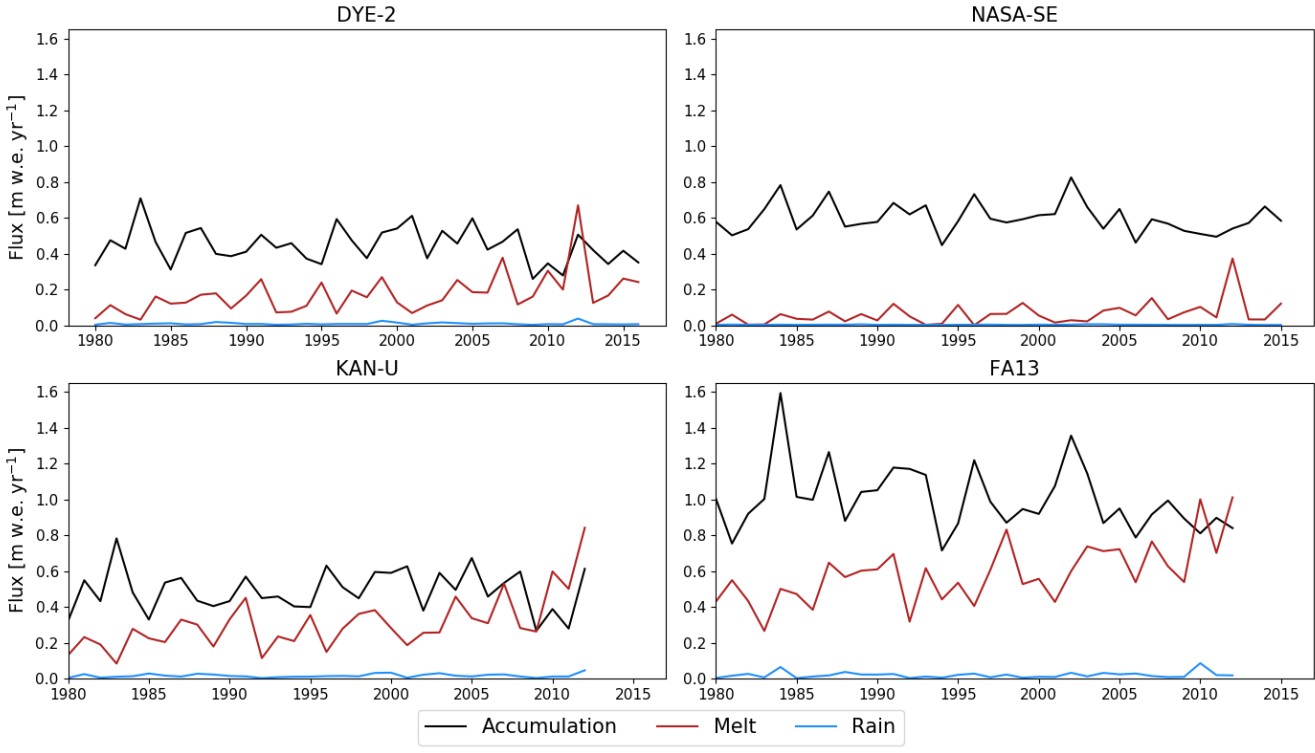

**Figure 2.** Annual surface mass fluxes from RACMO2.3p2 at the study sites (1980-drilling date)





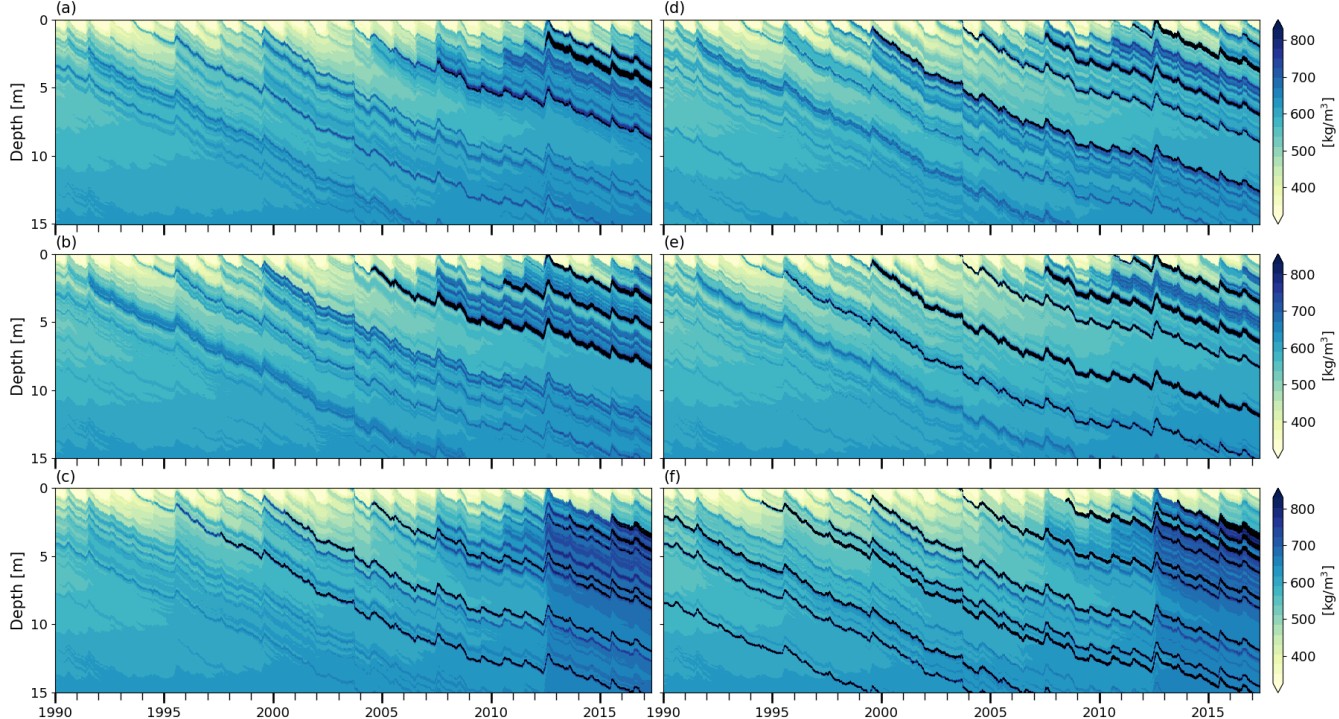

**Figure 3. Modeled firn density at DYE-2 (a) BK wh02 ip810, (b) R1M grLK ip810, (c) DPM grLK ip810, (d) BK whCL ip810, (e) R1M grA ip810, (f) DPM grA ip810, black indicates ice layers**



**Figure 4.** Measured and modelled depth-density profiles at DYE-2 on 11/05/2017. The modelled densities are averaged at the vertical resolution of the drilled core. CR: CROCUS, HL: Herron and Langway, KM: Kuipers Munneke.





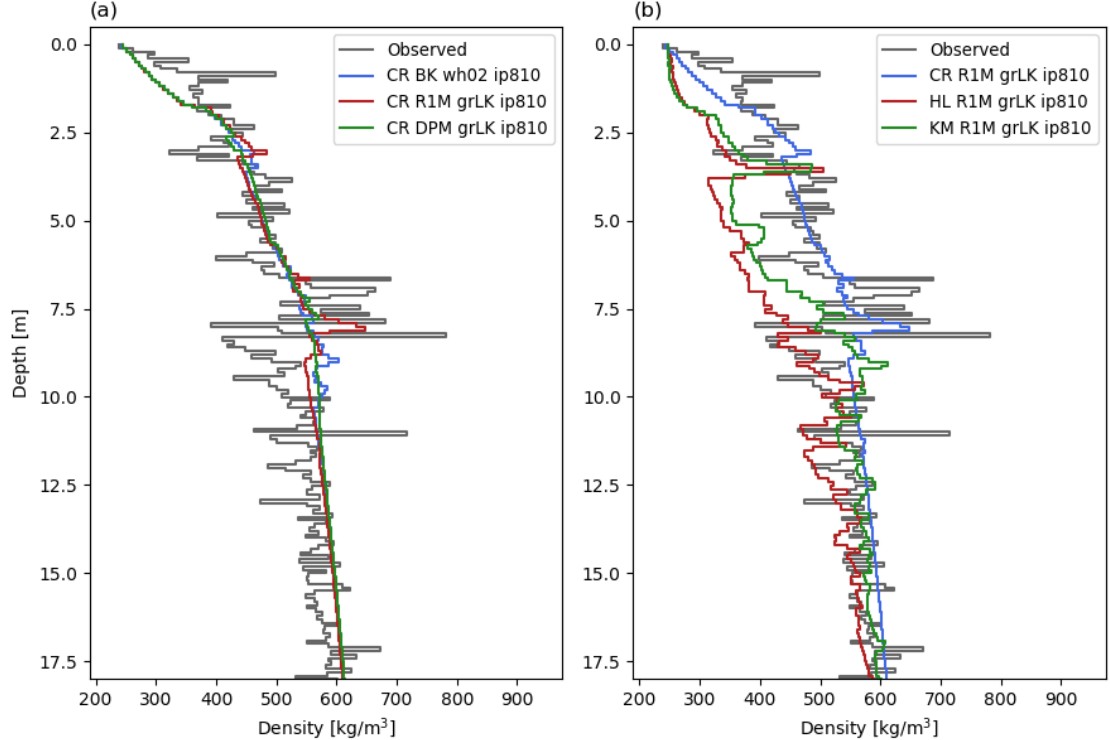

**Figure 5. Measured and modelled depth-density profiles at NASA-SE on 04/05/2016. The modelled densities are averaged at the vertical resolution of the drilled core. CR: CROCUS, HL: Herron and Langway, KM: Kuipers Munneke.**





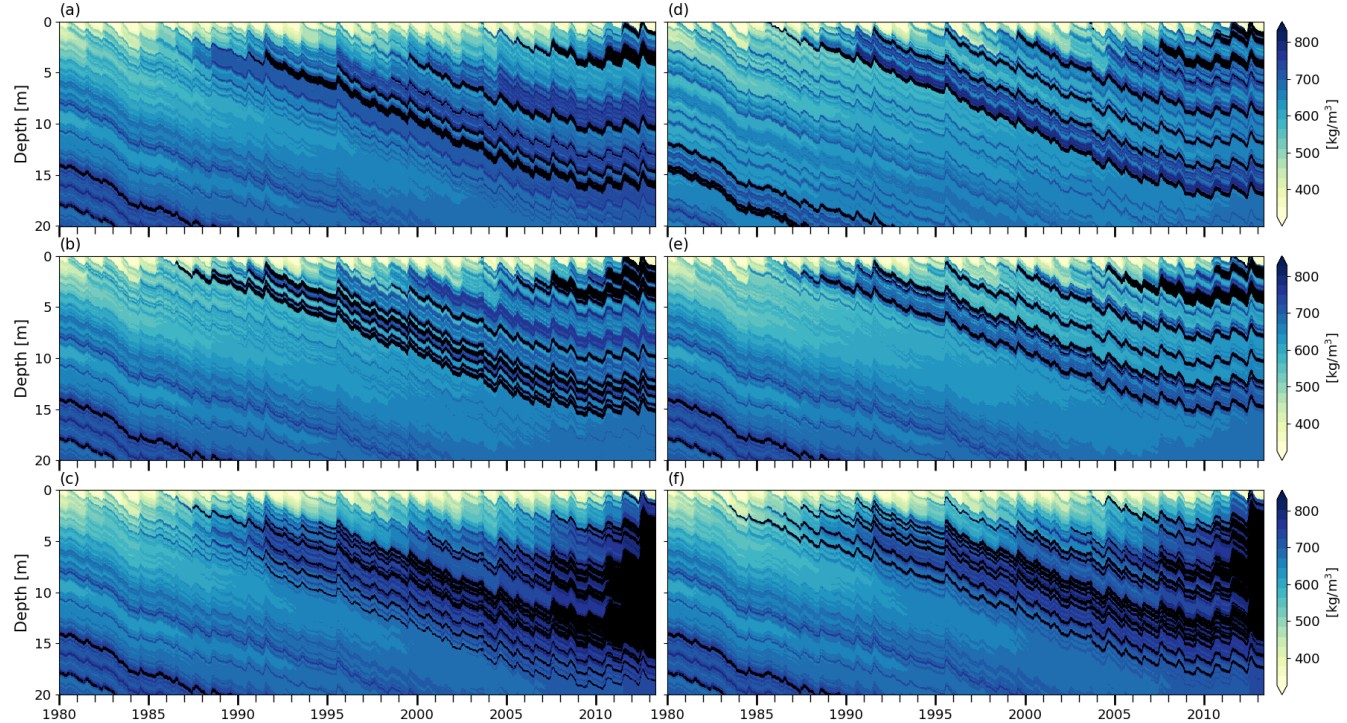

**Figure 6. Modelled firn density at KAN-U. (a) BK wh02 ip810, (b) R1M grLK ip810, (c) DPM grLK ip810, (d) BK whCL ip810, (e) R1M grA ip810, (f) DPM grA ip810, black indicates ice layers**

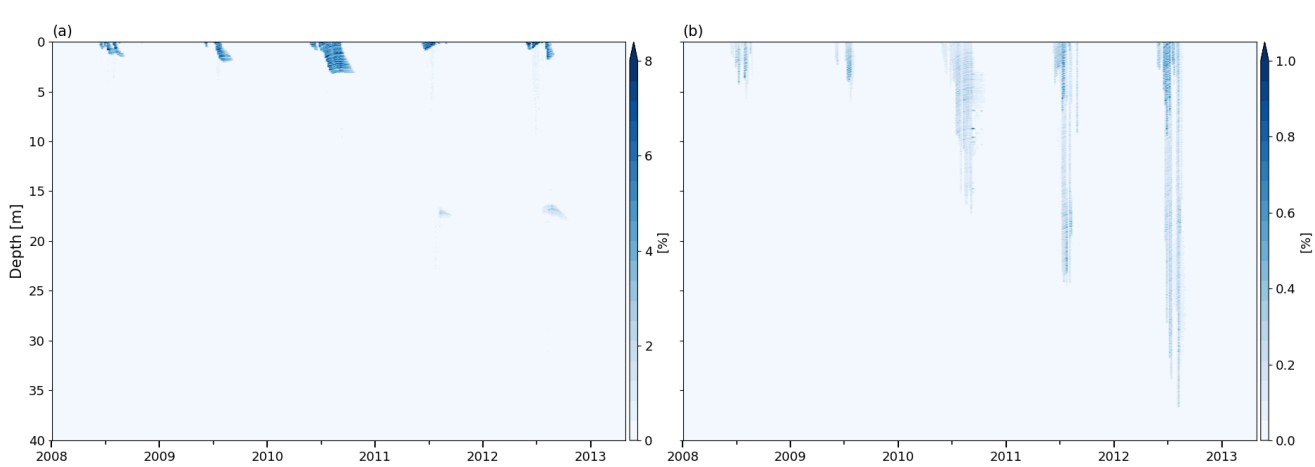

**Figure 7. Volumetric water content at KAN-U site for DPM grLK ip810 in (a) Matrix flow domain, (b) Preferential flow domain, notice the different scales**




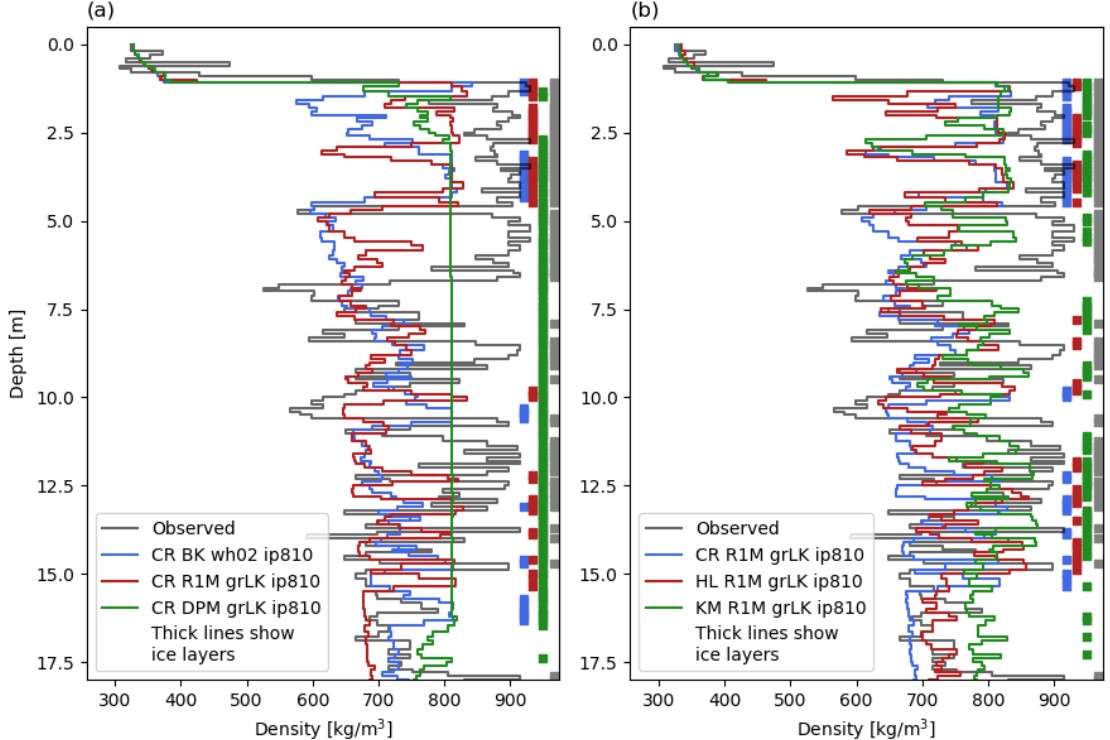

**Figure 8. Measured and modelled depth-density profiles at KAN-U on 28/04/2013. The modelled densities are averaged at the vertical resolution of the drilled core. CR: CROCUS, HL: Herron and Langway, KM: Kuipers Munneke.**



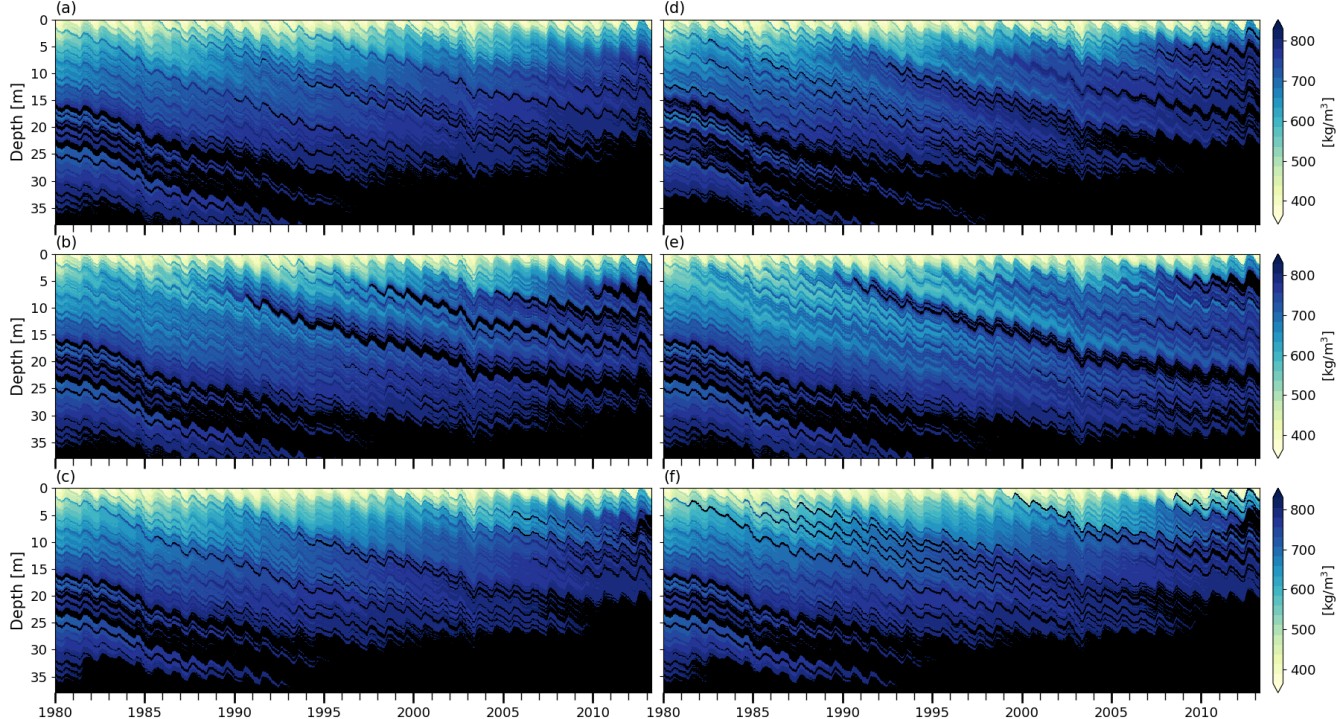

**Figure 9.** Firn density at FA13 (a) BK wh02 ip810, (b) R1M grLK ip810, (c) DPM grLK ip810, (d) BK whCL ip810, (e) R1M grA ip810, (f) DPM grA ip810, black indicates ice layers





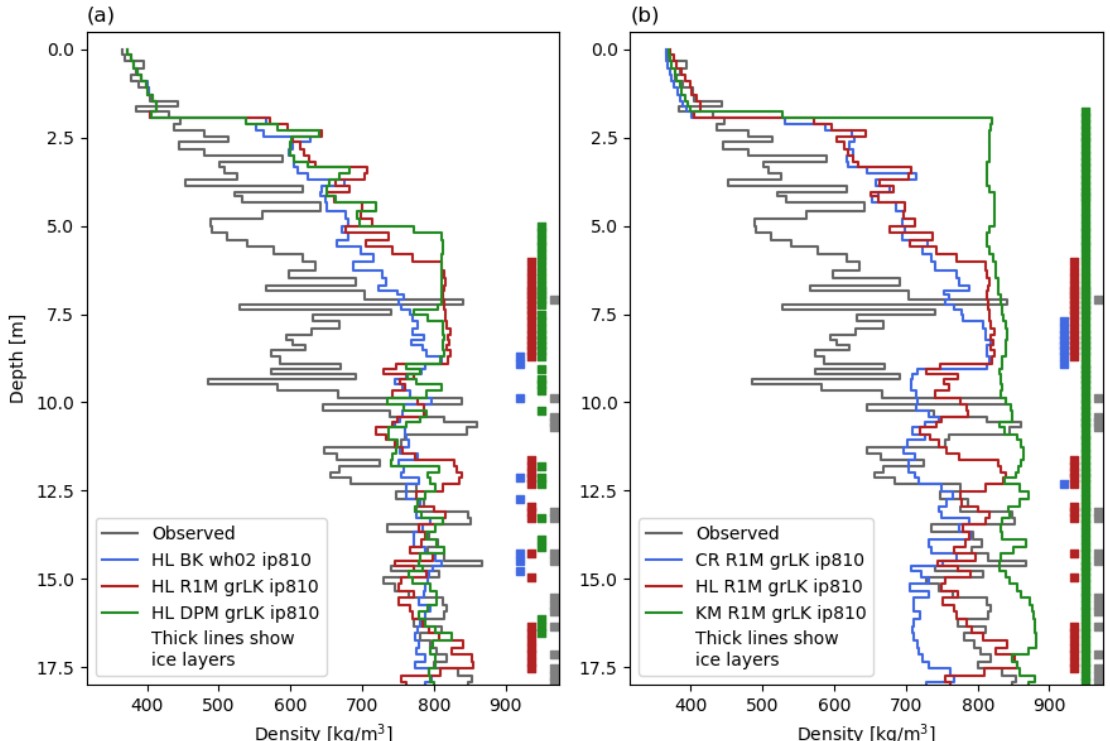

**Figure 10. Measured and modelled depth-density profiles at FA13 on 10/04/2013. The modelled densities are averaged at the vertical resolution of the drilled core. CR: CROCUS, HL: Herron and Langway, KM: Kuipers Munneke.**





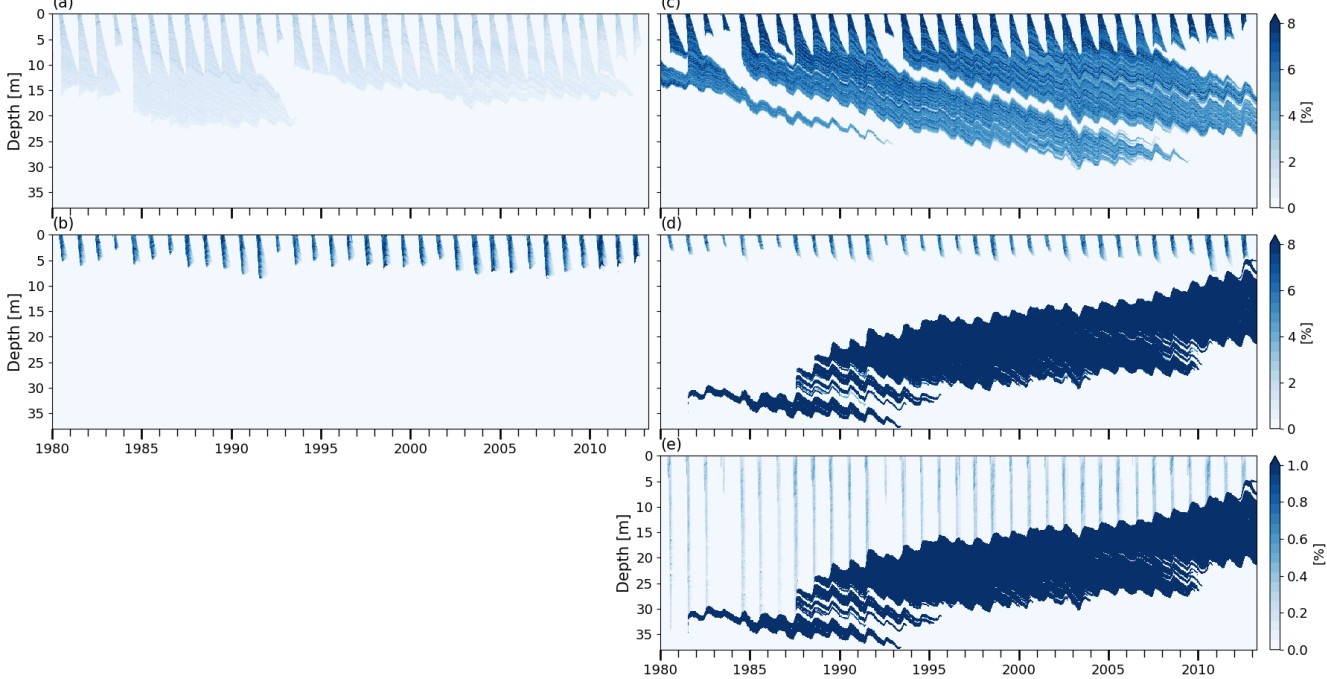

**Figure 11. Volumetric water content at FA13 (a) BK wh02 ip810, (b) R1M grLK ip810, (c) BK whCL ip810, (d) Matrix flow domain of DPM grLK ip810, (e) Preferential flow domain of DPM grA ip810, notice the different scales**

