# Peer review of "Development of physically based liquid water schemes for Greenland firn-densification models"

_The Cryosphere, 2019_

## Referee Comment (RC1) · Anonymous Referee #1 · 4 Mar 2019

[10pt,a4paper] article

graphicx latexsym [usenames]color amssymb

**Development of physically based liquid water schemes for Greenland firn-densification models. Verjans et al.**

[Figure]

**1 General Comments**

This paper makes an extremely valuable contribution to our understanding of the performance of models developed to describe the densification of snow in the percolation zone. These models are important not only for the interpretation of elevation change in terms of mass balance change, but also in the calculation of meltwater output from ice sheets. The authors compare 3 methods of modelling the movement of water in firn; a simple bucket model, a one-dimensional Richards Equation model for flow in porous media and a combination of porous flow and faster flow through preferential pathways. These models are combined with 3 different methods of calculating densification; the linear viscous model used in the CROCUS model, the Herron and Langway model for dry snow densification and the Kuipers-Munnecke model. Input data comes from the RACMO2 regional climate model, available at 5.5 km horizontal resolution at 3-hourly intervals.

The authors choose input data from 4 different locations on the Greenland Ice Sheet and compare the model output to measured firn density profiles from single cores collected at these locations. Although it is clear they are aware of the extreme variability of firn density in the percolation zone, they do not comment on the difficulty of up-scaling from a single point measurement to the model scale. It would be very useful to have a discussion, early on in the paper, about what features of the observed data a good model, fed by their input data, could be expected to reproduce. Perhaps the overall densification rate? Perhaps the amount of refrozen meltwater in the profile? Certainly not the position and size of ice layers as this is notoriously variable.

Given this preliminary discussion the authors would be able to concentrate on the differences in model outputs, which are interesting and illuminating, and could avoid putting too much weight on the comparison with observed data. Small changes would

probably be enough; for example, if Figures 4, 5, 8 and 10 had a very pale grey indication of the observed data in the background, the model outputs would be easier to compare with each other and the reader would be led away from the idea that the model output "should" reproduce the point observations in detail.

**2   Detailed comments**

The descriptions of what each model is doing are very good and show great insight into how these quite complex models work. However, the mass of material in the Results section is rather overwhelming and the authors might consider whether all of it is needed. It may be that all the results are indeed required, in order to justify the points made in the Discussion section. In that case, maybe the answer would be to move some material into a Supplementary Material section.

The statement (p.8 l.26) that water cannot fill the entire pore space because otherwise there would not be enough space for the liquid to freeze is a bit confusing. Is it equivalent to saying that as water in a saturated layer freezes, the part that won't fit is expelled into another layer? What about the possibility that a saturated layer expands on freezing?

**3   Technical Comments**

The paper is generally well written in very good English. Just occasionally the first author has been deceived by "false friends" . It would be worthwhile looking at each use of "therefore" to ensure that the authors really do mean a strong causal connection; the alternative might be to use the weaker 'so" or even "and".

- p. 1 l.22 "in actuality" should be "in reality"

- p.2 l. 29 "through the porosity" should be "through the pores"

- p.3 l.2 "Snow models have developed" should be "Snow modellers have developed"

- p.4 equation(2) and other equations following. If the numerical value of a parameter is given in an equation it should have its units attached e.g. 273.15K. Otherwise use parameter names and define them eg $\eta_0$

- p.5 l.14 lower case rather than upper case for " Where"

- p.5 l.15 If $r_0$ has units of m then so must $b_0$. $b_1$ has units of m $K^{-1}$ and $b_2$ m yr/m w.e.

- p.5 equation (8) Roman font for "exp"

- p.6 l.26 "the depth from which firn density does not reach"? Not clear what is meant here

- p.8 l.6 and elsewhere. "capillary suction" should be "capillary tension"

- p.9 l.7 "superior to zero" should be "greater than zero"

- p.9 l. 23 "above the impermeability threshold"? Not clear what is meant here.

- p.11 l. 16 "synthetically" should be " artificially"'

- p.25 l.19 maybe say "developed for seasonal snow models"?

---

## Referee Comment (RC2) · Baptiste Vandecrux (Referee) · 13 Mar 2019

**Review of *Development of physically based liquid water schemes for Greenland firn-densification models* by Verjans et al.**

Baptiste Vandecrux (bava@byg.dtu.dk)

This study investigates different formulations of liquid water routing, firn densification and grain growth within the community firn model. Various configurations are tested to investigate the sensitivity of the models' internal parameters. The results from each model version are compared and validated against in situ density and temperature observations. The study evaluates whether more complex modelling approaches improve the comparison with observations and identifies the parameters and processes that should be better constrained and/or developed in future firn models.

Firn modelling is important for surface mass balance calculation, altimetry correction, ice core interpretation and the study of meltwater movement is a great challenge for the firn community. The manuscript presents, to my knowledge, the first application of a firn model including heterogeneous meltwater percolation on the Greenland ice sheet. The testing protocol is robust and the entire manuscript is nicely written and well structured. The complex model setups and results are clearly presented. The manuscript is without doubt worth publishing in the Cryosphere after the following comments are addressed.

1) The authors selected from the SUMup dataset few density profiles for validation, but left aside other profiles contained in that same dataset. Some of these missing measurements can be deemed redundant, but other firn observations, such as the PARCA cores, are important to understand the performance of the model through time.

2) Another flaw of the manuscript is the lack of context regarding the modelling approaches used here as well as the absence of comparison of the results with previous similar work. It is important to know if the results presented here were already known or expected or if they constitute a novelty of this study.

3) Some of the reasons that could explain the model performance, such as potential biases in the forcing data or the effect of the impermeability threshold on the calculated FAC, are not even mentioned. These sources of bias should be quantified and discussed.

Specific comments and suggestions are reported below.

[revised manuscript text omitted]

---

## Referee Comment (RC3) · Anonymous Referee #3 · 26 Mar 2019

In this paper, authors simulated liquid water infiltration and densification of firn using several model schemes. They were performed for four field sites in Greenland. Validation using the field observation data was also performed. In my opinion, this study provides scientific valuable results. I have several suggestions to provide better information.

1) Measured temperature was used for the validation of models. Simulated runoff was also discussed. However, figures for these results were not shown. Lack of figures sometimes makes difficult to understand in detail. In TC, authors can use a supplement file to show figures relatively less important. Therefore, figures of them should

be added using a supplement to support the discussion in the main text (see minor comment P13 L22-28, P14 L14-15).

2) The simulation in this paper was performed for four fields. Tables were provided for results in each field showing total or averaged simulated values. I would like to suggest that the author provide a table which shows the comprehensive result to see the difference between fields about simulation results (see minor comment in P19L5 and P22L26). This is not prerequisite for acceptance, but it will help to understand the overall result.

3) Although the main target of this study was the validation of liquid water infiltration model, density and temperature data were used instead of liquid water content for validation. It leads to the limitation of the validation itself. Discussion about the limitation because of this is also necessary.

4) This simulation study can provide several suggestions for a laboratory experiment and field observation required to improve the model. Although limitations are written in conclusion, detailed discussion about limitation and suggestion of new experiment and observation (e.g. liquid water infiltration experiment into firn or observation of liquid water.) will be informative for future research.

Minor comments

P8 L26-27 In this sentence and Eq. (17), the saturated water content was estimated as i/w (0.917?) of pore space. It seems to be used for convenience in calculation. Yamaguchi et al. (2010) obtained the saturated water content was about 90% of pore space in their gravity drainage column experiment. Although they are coincidence, this paper had better be referred to show that the assumption in Eq. (17) is consistent.

P13 L22-28 Figures for simulated runoff had better to be shown in a supplement.

P14 L14-15 Comparison between observation and simulation of temperature also needs a figure in a supplement.

P17 L18-30 In terms of average value, DPM had a good agreement. However, the simulated result of DPM was a constant value for vertical and maximum density was underestimated. In my opinion, the depiction "good agreement" feels not suitable (depiction "average density is reasonable" may be OK). The state of this result had better be that present liquid water infiltration scheme has limitation and requires future improvement.

P19L5 Here, densification model had a larger effect than water infiltration model. Is this trend only this place or common for all four places? Comprehensive table comparing between fields about this will help to check this question. (This is suggested in major comments.)

P20 L27 L31 Fig8f? Recheck the figure number.

P22 L7-9 Is it mean that the simulation using BK and R1M were performed by CROCUS whereas DPM was performed by SNOWPACK? If so, the difference in results receives the effect of the difference of numerical snowpack model. Did authors check the difference of them performing a simulation with the same water infiltration scheme? SNOWPACK has Bucket and RE scheme comparable with BK and R1M, respectively.

P22 L26 This results seems that the DPM is not suitable for this place (actually, improvement is necessary for preferential flow scheme). As discussed in P17 L4-8, BK and R1M reproduce surface ponding and refreeze. If a suitable model is different depending on the field, the most suitable scheme had better be shown for each field. Comprehensive table suggested in a major comment is useful to show a comparison between fields.

---

## Author Comment (AC1) · 3 May 2019

Dear Dr. Karlsson,

We would like to thank you for your attention to our manuscript and the three reviewers for their comments. We are pleased that all three reviewers liked the manuscript and consider it to be publishable in 'The Cryosphere'. We appreciate both their interest in our study, and their constructive comments, and we have paid great attention to the feedbacks provided in revising our manuscript.

We have provided a response to each reviewer, including details of any edits to the manuscript. Minor text edits are detailed in these responses and the more substantive revisions consist of:

1) We have clarified our methodology and the conclusions we draw from our results.

2) We provide a Supplementary Information file which contains additional tables and figures, as requested by the reviewers, to help in the interpretation of the results.

Note that all the material included in the Appendix of the original version of the manuscript has also been moved in this Supplementary Information file.

Please do not hesitate to get in touch should any further modifications be required, or any clarification on our response to reviewers.

Best regards,
Vincent Verjans, on behalf of authors

**Referee 1**

We thank this reviewer for their compliments and feedback. Our response to these comments is below, the referees original text is given in red throughout and our response is in black.

A) General comments

Although it is clear they are aware of the extreme variability of firn density in the percolation zone, they do not comment on the difficulty of up-scaling from a single point measurement to the model scale. It would be very useful to have a discussion, early on in the paper, about what features of the observed data a good model, fed by their input data, could be expected to reproduce. Perhaps the overall densification rate? Perhaps the amount of refrozen meltwater in the profile? Certainly not the position and size of ice layers as this is notoriously variable.

We recognize that one would not expect a firn model to reproduce with great precision the number of ice layers observed and their exact location, especially given the spatial variability of firn structure, the 5.5 km resolution of the Regional Climate Model used as forcing and the use of a constant surface density. However, we think that the assessment of the ability of a firn model to produce *representative* ice layers and a density profile varying in depth is valuable for some applications, especially for the meltwater routing.

In the manuscript, we try to highlight the ability of the models tested to match observed values of FAC and 10 m temperature. These variables give, in our opinion, a good perspective on how well a particular model reproduces the bulk conditions of the upper firn column.

As you suggested, we added a paragraph at the beginning of Sect. 4 in order to clarify the way in which we compare model results to observations to the reader.
Section 4: *Results of simulations and observations are inter-compared based on the firn air content (FAC; the depth integrated porosity in a firn column) over the top 15 m of firn and the temperature at 10m depth. Comparing the modelled FAC and 10 m depth temperature values with observed data depicts the ability of the tested models to reproduce the bulk condition of the upper firn column. We also qualitatively assess the degree to which the models to form a 'realistic' ice layer distribution and depth-density profile. One would not expect simulated values of either to match observations precisely given the high spatial variability of firn structure (Marchenko et al., 2017), but it is indicative of the models' performance in reproducing heterogeneity in firn density.*

Given this preliminary discussion the authors would be able to concentrate on the differences in model outputs, which are interesting and illuminating, and could avoid putting too much weight on the comparison with observed data. Small changes would probably be enough; for example, if Figures 4, 5, 8 and 10 had a very pale grey indication of the observed data in the background, the model outputs would be easier to compare with each other and the reader would be led away from the idea that the model output "should" reproduce the point observations in detail.

In our paper we do not aim to single out a specific water flow scheme or model configuration as "best" in comparison with observations. We focus on giving the reader an extensive overview of the behaviour and differences in outputs of the three schemes tested. We hope that the modifications made in response to your comment above, amongst others, addresses this comment.

We additionally changed Figures 4, 5, 8 and 10 as you suggested by giving a pale grey colour to the observed depth-density profile.

The descriptions of what each model is doing are very good and show great insight into how these quite complex models work. However, the mass of material in the Results section is rather overwhelming and the authors might consider whether all of it is needed. It may be that all the results are indeed required, in order to justify the points made in the Discussion section. In that case, maybe the answer would be to move some material into a Supplementary Material section.
We agree with you that the manuscript is long. We have considered to move parts of the Results section to the Supplementary Material. However, we think that the clarity and coherence of the study would be affected by any such move and we took the decision to keep the entirety of the Results section in the main manuscript.

The statement (p.8 l.26) that water cannot fill the entire pore space because otherwise there would not be enough space for the liquid to freeze is a bit confusing. Is it equivalent to saying that as water in a saturated layer freezes, the part that won't fit is expelled into another layer? What about the possibility that a saturated layer expands on freezing?

Our approach in this study has been to implement the SNOWPACK liquid water schemes with the greatest fidelity possible. The deviations we make from the original schemes are only related to the age and depth scales that differ by orders of magnitude between seasonal snow and firn. Because the SNOWPACK scheme uses the correction for the saturation that you refer to, we have decided to implement the same. Please note that this correction (that, at saturation, water does not fill the entire porosity of the firn) is in good agreement with the laboratory findings of Yamaguchi et al. (2010).

We add a statement to this effect in the revised manuscript in Sect. 3.2:

*It is reasonable to use this correction factor since Yamaguchi et al. (2010) found that trapped air still occupies 10 % of the porosity in saturated snow.*

It would be worthwhile looking at each use of "therefore" to ensure that the authors really do mean a strong causal connection; the alternative might be to use the weaker 'so' or even "and".

In the revised manuscript we looked again at our uses of "therefore" and removed or replaced several ones.

B) Specific comments

For all the specific editorial comments, we agreed with the reviewer and made modifications as suggested.

Concerning the comment:

p.6 l.26 "the depth from which firn density does not reach"? Not clear what is meant here

We replaced the statement by "the depth below which all layers have density higher than the pore close-off value (830 kg m$^{-3}$)", as suggested by another referee.

**Referee 2**

We thank this reviewer for his compliments and feedback. Our response to his comments is below, the referees original text is given in red throughout and our response is in black.

General comments

The authors selected from the SUMup dataset few density profiles for validation, but left aside other profiles contained in that same dataset. Some of these missing measurements can be deemed redundant, but other firn observations, such as the PARCA cores, are important to understand the performance of the model through time.

We greatly value the existence of the SUMup and PARCA datasets because they enable an extensive evaluation of firn densification modelling against observations in general. These are thus very valuable for any firn modelling study at the ice-sheet wide scale.

Since we, to our knowledge, are the first study to perform firn densification simulations using these more physically complex liquid water flow schemes, we preferred to focus our analysis on the performance of the different schemes at four sites chosen on the basis that they represented a range of climatological and glaciological settings. Including more sites would have necessitated a reduction in the detail of our analysis which would have had a detrimental effect on its thoroughness. Reviewer 1 in fact complimented the study in this respect.

We agree that the performance of the model through time is important, but given the current length of the manuscript we feel that an assessment of this is beyond the scope of this study. We will certainly consider this in future work however and have added a comment in the Conclusion of the revised manuscript to this effect.

Another flaw of the manuscript is the lack of context regarding the modelling approaches used here as well as the absence of comparison of the results with previous similar work. It is important to know if the results presented here were already known or expected or if they constitute a novelty of this study.

We thank the reviewer for bringing these papers to our attention. In this revised version, we took care to refer more to both existing modelling studies and observational studies as you advised. For example, we added references to Langen et al. (2017) and Steger et al. (2017b), in which model experiments were performed at KAN-U with different firn models/liquid water schemes/climatic forcing. We discuss the differences with our results in section 5. We also included references to Marchenko et al. (2017) who highlighted the large spatial variability of firn structure and to Miller et al. (2018) who measured discharge rates in firn aquifers. We performed further simulations with runoff rates in aquifers calibrated in accordance with the observations of Miller et al. (2018) and discuss these briefly. We think these additions improve the quality of the manuscript.

References to Marchenko et al. (2017)
Section 4: *We also qualitatively assess the degree to which the models to form a 'realistic' ice layer distribution and depth-density profile. One would not expect simulated values of either to match observations precisely given the high spatial variability of firn structure (Marchenko et al., 2017), but it is indicative of the models' performance in reproducing heterogeneity in firn density.*

Section 5: *On the other hand, comparisons between modelled and observed density profiles are strongly affected by the choice and accuracy of the densification formulation, the variability of surface density, several other factors influencing model outputs mentioned above and possible uncertainties in field measurements. Such uncertainties are related to the strong spatial variability of firn structure (Marchenko et al., 2017), which can be observed by comparing density profiles of cores drilled at nearby locations.*

Reference to Miller et al. (2018)
Section 3.4.2: *Miller et al. (2018) found discharge rates within the firn aquifer to be 4.3 x 10$^{-6}$ m s$^{-1}$ by borehole dilution tests in the field. We tested this approach in our model by applying this value as a constant discharge rate for aquifers formed in our simulations. We found however that, using this approach, an aquifer was not sustained; suggesting that such discharge rates must be dependent on the total amount of water within the aquifer and are likely temporally variable.*

Reference to Langen et al. (2017) and Steger et al. (2017b)
Section 5: *We compare the results reached at KAN-U with previous firn modelling studies at this site. Langen et al. (2017) also used the CROCUS densification scheme and a water flow scheme conceptually comparable to R1M but with a simplified solving process of the RE. Their results show a density profile entirely at ice density from 4 m depth and an overestimation of the temperature at 10 m depth (+3 to +5 K). In our study, the R1M results show lower deep densities and a 10 m temperature underestimation. These discrepancies can be attributed to differences in 1) model implementation, 2) climatic forcing and 3) details of the water flow scheme. Model resolution is likely of importance here: Langen et al. (2017) used a coarser vertical resolution, disfavouring the formation of thin impermeable layers, allowing liquid water to flow more readily to greater depths and refreeze, thus causing greater density and higher temperature values. Steger et al. (2017b) used the SNOWPACK densification model (recall: we use different densification physics to SNOWPACK in this study) with a bucket scheme, similar to BK, and the same climatic forcing as in this study. Their model output shows a firn column fully compacted to ice at KAN-U. They attribute this to the overestimation of densification rates by the densification model used and also argued that some other effects*

*could be at play such as the surface density applied. The 10 m temperature is underestimated in their result, likely due to the absence of percolation of water through ice and thus no latent heat release at depth.*

Some of the reasons that could explain the model performance, such as potential biases in the forcing data or the effect of the impermeability threshold on the calculated FAC, are not even mentioned. These sources of bias should be quantified and discussed.

Concerning the RCM forcing data, we briefly mention that any bias in the climatic forcing would affect model results in Sect. 5.

We do not discuss this issue of RCM validation and uncertainty in-depth in the manuscript because we consider that the main goal of this study is to investigate the impact of different liquid water schemes on model performance. At the present state, we consider the biases arising from the liquid water scheme and the densification model as much greater than the uncertainty arising from the RCM. We think that discussing the validity of the climate model is beyond the scope of this study and such discussions are extensively covered in the existing literature.

We added additional references to Noël et al. (2018) and Ligtenberg et al. (2018) in Sect. 2.2 of the revised version of the manuscript. These papers evaluate RACMO2.3p2 and assess the effects of recent developments of the RCM on firn models respectively. Please note that the biases reported in Noël et al. (2018) are further reduced in our study because we use the RCM at a 5.5 km resolution. In Noël et al. (2018), there are detailed comparison between modelled and observed values of snowfall (see Fig. 11) and the estimated uncertainty integrated over the GrIS and specific catchments is 10 %. Also, it is shown (Figs 5 and 10b) that melt rates are well reproduced along the K-transect and at a South-East Greenland site, which are areas corresponding to the four sites of model experiments in our study. In Noël et al. (2018), radiative fluxes, which are the primary drivers of melt rates, are also demonstrated to be in good agreement with measurements from the PROMICE weather stations (Fig. 4) and their validation is further detailed in the supplementary material.

Section 2.2: *We refer to Noël et al. (2018) for a detailed discussion about the performance of RACMO2.3p2 on GrIS and related uncertainties. Additionally, Ligtenberg et al. (2018) have demonstrated the impact of recent developments in RACMO2.3p2 on firn modelling, mostly yielding improvements in modelled densification.*

Considering the impact of the impermeability threshold on the FAC, this is one of the main point of investigation of this study. At every site, we vary the impermeability threshold of each of the flow scheme and quantify the change in FAC due to these variations. The relative changes of FAC between simulations are given in the text and the absolute values as well as their discrepancy with the observed FAC are provided in the Tables. We also try to provide explanations of why an increase/decrease of the impermeability threshold can cause a modification of the modelled FAC.

We recognize that the discussion about the DPM results at KAN-U in Sect. 5 was speculative, in part because the prescribed impermeability threshold was limiting the amount of refreezing allowed in individual layers. Therefore, we have decided to remove this specific part of the text (p.22 l.23-30 in the original manuscript).

Specific comments

p.1 l.13 The abstract could benefit from being more quantitative and more specific.
We added quantifications of the increase in melt over the GrIS and an order of magnitude of the changes in modelled firn densities due to the use of different flow schemes. We specified the main uncertainties affecting the performance of the water flow schemes implemented. We also specified the processes that firn models should better represent for reaching improved results.

p.1 l.17 I believe the appropriate word is just "bucket approach". "tipping" gives the idea that the bucket/layer is emptied into the next bucket/layer. The water rather fills the layer's capacity and the excess water overflows into the next layer.
We agree with this and have replaced "tipping bucket approach" by "bucket approach" in the entire text.

p.1 l.21 Quantify
We proceeded to the suggested modifications: we quantify the trend of increase in surface melt and the upper limit of inter-model differences of porosity that we reach in our simulations.

p.1 l.26 Please list the main ones.
We proceeded to the suggested modification: we give as examples the snow grain structure and the presence of ice layers.

p.1 l.28 Name them.
We proceeded to the suggested modification by naming the microstructural effects, the wet snow metamorphism and the temperature sensitivity.

p.2 l.10 Please clarify this sentence and give reference.
We could not find any reference to this sentence and so have removed it from the revised manuscript.

p.2 l.29 Maybe "through the pores" ?

We proceeded to the change suggested.

 Maybe implemented or combined with?
We proceeded to the change suggested.

 It would be useful to know how the HL and KM densification schemes are forced. Are the accumulation and temperature considered constant?

HL and KM are forced with exactly the same three hourly climatic forcing as CROCUS. In the KM model, the firn temperature is used instead of the annual mean temperature for the diffusion process, as proposed in Steger et al. (2017). For the grain growth process, we use the 10 m depth temperature instead of the annual mean because the release of latent heat is not captured by the latter value. For the HL model, we use the firn temperature instead of the annual mean as in Lundin et al. (2017). This allows for seasonal variability in densification rates and accounts for latent heat release. For both the HL and KM, we use the mean accumulation rate over the lifetime of any firn layer instead of the mean annual accumulation rate (see Stevens 2018) because this approximates the overburden stress better.

These information are added in Sect. 4.1 in the revised manuscript.

 Please give these values.
We added the specification of the surface density values in Sect. 2.4.

 More density profiles are available at NASA-SE and DYE-2 from the PARCA campaign. Considering that they are freely accessible in the SUMup dataset, they should be included in the comparison. More measurements are available at KAN_U in 2015, 16, 17. Considering that these surveys are close in time, it might be deemed unnecessary to add them to the comparison. But they should be mentioned and leaving available data out should be justified. I believe there is also a density profile from the firn aquifer location from 2016.

We addressed the issue of simulating firn density at more locations and proceeding to comparisons with more observations in the response to your general comments. Please note that we took care to justify our approach at the end of Sect. 2.4.

 I am missing a description of the temperature measurement that is being used in the validation. Here would be the appropriate place to describe them.

We added details of the temperature measurement used at KAN-U in Sect. 2.4 of the revised manuscript. At all the other sites, the temperature measurement comes with the source cited for the density data. We specify this also in Sect. 2.4 of the revised manuscript.

 Maybe: ... is the depth below which all layers have density higher than the pore close off density.
We proceeded to the change suggested.

 Please refer to section 2.4.2 where lateral runoff is detailed.
The BK model does not use the same runoff routine as the R1M and DPM. In the BK, all the water percolating until an impermeable ice layer is treated as immediate lateral runoff. We modified the manuscript in Sect. 3.1 in order to clarify this point.

 theta and its unit are actually not defined clearly in a sentence.
$\theta$ and its units are defined in p.4 Eq.(3). We added a sentence in the manuscript to refer the reader back to this equation for the definition of $\theta$.

 maybe "tuning" coefficients?
We proceeded to the modification suggested.

 This value of irreducible water content was used in
https://doi.org/10.5194/tc-6-743-2012 and https://doi.org/10.3389/feart.2017.00003 where it was described as:
"The irreducible water content is set to a relatively low value of 2% of the pore volume to mimic processes that allow an effective vertical water transport to lower layers, such as piping"

Here you use that same value for a model that solves these processes explicitly and therefore a low irreducible water content is questionable. Why not using the Coleou and Lesaffre formulation?

Throughout the entire implementation of R1M and DPM, we have taken care to remain as close as possible to the original SNOWPACK implementation. In the SNOWPACK implementation, the value of 0.02 is used for $\theta_r$ instead of the Coléou and Lesaffre formulation.

We would like to point out the difference between the water holding capacity $\theta_h$ of the BK and the residual water content $\theta_r$ of the R1m and DPM. $\theta_h$ is meant to simulate retention of water by capillary tension against gravity. The capillary tension is however explicitly modelled when applying the Richards Equation. The residual water content refers to the saturation at which the

mobile water loses its continuity in the porous medium, and water flow becomes impossible (see for example Szymkiewicz (2009)). Thus, the water holding capacity should represent the effects of both this residual water saturation and of the capillary forces exerted when firn effective saturation is greater than zero (that is, when the volumetric water content is greater than the residual water content).

p.9 l.7 Is the following related to the sentence above: "... but in case of refreezing, can approach zero and must be adjusted accordingly"? If yes, then the word "another" is misleading. If not then please develop the refreezing issue as the provided information is insufficient to understand why an adjustment is necessary.

We agree with your comment and have modified this paragraph accordingly. We hope that it now provides better clarity.

p.9 l.9 In theory, water content below irreducible water content should not suffer gravity as capillary forces dominate. It is also unclear why changing from 0.75 to 0.9 addresses this issue. Please clarify.

We agree that in theory, the value of $\theta$ should be allowed to equal and to remain at a fixed $\theta_r$ value. However, the numerical requirement of keeping $Se$ strictly greater than zero leads the model to adjust $\theta_r$ when $\theta$ approaches $\theta_r$ and minimal flow rates thus persist.

We clarify this point in Sect. 3.2 of the revised manuscript.

Note that gravity-driven flow occurs in the model as a function of the hydraulic conductivity, which depends on the effective saturation. Since the effective saturation must remain above zero, very low flow rates persist in layers close to $\theta_r$. In other words, because $Se$ cannot be set to zero, extremely low rates of water flow are modelled, and over long periods of time (weeks or months) this leads any layer to fall below the threshold we use to consider the firn column dry. The use of 0.9 instead of 0.75 leads to higher values of $\theta_r$ when $\theta$ approaches zero. In turn, this causes the effective saturation value to be lower. Lower effective saturation values lead to lower hydraulic conductivity and thus decreased flow rates.

p.9 l.23 Please point at section 2.4.2 for runoff calculation.
We proceeded to the modification suggested.

p.11 l.15 Please compare this number to field observations:
Schneebeli, M.: Development and stability of preferential flow paths in a layered snowpack, in: Biogeochemistry of Seasonally Snow-Covered Catchments (Proceedings of a Boulder Symposium July 1995), edited by: Tonnessen, K., Williams, M., and Tranter, M., 89–95, AHS Publ. no. 228, 1995.

Marsh, P. and Woo, M.-K.: Meltwater movement in natural heterogeneous snow covers, Water Resour. Res., 21, 1710–1716, doi:10.1029/WR021i011p01710, 1985.

Williams et al. Visualizing meltwater flow through snow at the centimetre-to-metre scale using a snow guillotine *Hydrol. Process.* **24**, 2098–2110 (2010)

The parameter $\Theta$ is the threshold saturation value in the preferential flow domain at which point water is assumed to flow back to the matrix flow domain. This is purely used as a tuning coefficient to reach the best agreement possible between modelled and observed ice layer formation and runoff rates by Wever et al. (2016) and $\Theta$ is difficult to compare to any observation.

The parameter $N$ should not be confused with the parameter $F$. The latter represent the proportion of the snowpack where preferential flow can occur. As explained at the beginning of Sect. 3.3, we do not follow the regression relation derived in Wever et al. (2016) for this parameter but we take a constant value of 0.2 in line with observations of Marsh and Woo (1984) and Williams et al. (2010). The parameter $N$ represents the number of active flowpaths for preferential flow. As an example in a 1 m$^{-2}$ section of a snowpack and using $F = 0.2$, a single flowpath of area 0.2 m$^{-2}$ would correspond to $N = 1$ whereas 10 flowpaths of area 0.02 m$^{-2}$ would correspond to $N = 10$. Wever et al. (2016) briefly discuss the difficulty to constrain the $N$ parameter with observations: it is possible to detect the number of flowpaths that formed in a dye tracing experiment but it is much more difficult to evaluate how many flowpaths were active simultaneously. As a consequence, similarly to what has been done for $\Theta$, they proceeded to the calibration of $N$ by minimizing the mismatch between the final output of the model and their observations of runoff and ice layer formation in alpine snow.

p.11 l.24 Point at the section where the time step of the water scheme is introduced.
We proceeded to the modification suggested.

p.11 l.25 Unclear whether this sentence applies to both R1M and DPM or just to DPM.
We modified the manuscript to clarify this point.

p.12 l.3 Please discuss the chosen runoff rates and aquifer dynamics compared to
Miller, O., Solomon, D. K., Miège, C., Koenig, L., Forster, R., Schmerr, N., Montgomery, L. (2018). Direct evidence of meltwater flow within a firn aquifer in southeast Greenland. Geophysical Research Letters, 45, 207–215.
https://doi.org/10.1002/2017GL075707

We thank you for pointing us to this interesting and complementary observational study. We performed additional simulations by applying the runoff rates measured in that study in the simulated aquifers and discuss these new outputs and their implications at the end of Sect. 3.4.2.

p.12 l.4 Not in BK?
In this study, BK assumes instantaneous runoff when percolating water reaches an impermeable layer. Only the water held due to the water holding capacity can remain in firn layers. We clarified this point in Sect. 3.1 of the revised manuscript.

p.12 l.5 Isn't it allowed anywhere the water flow is impeded?
We have modified Sect. 3.4.2 to provide greater clarity.

p.12 l.10 The phrasing here and the structure of the section is rather confusing. It presents first the ponding on at the bottom of the column, then the lateral runoff that applies everywhere in the column, and finally reintroduce the ponding and show that the runoff rates do not allow the build up of aquifer. An alternative would be to present the runoff rates that apply on any ponding water in the column, then show that it does not allow the build up of aquifer, and finally present the aquifer-specific handling of ponding water as an exception.
As mentioned in the previous answer, we have modified Sect. 3.4.2 to provide greater clarity. We think that this concern is now addressed.

p.13 l.8 A major omission in this manuscript is the comparison of the forcing data to other estimations of accumulation, meltwater production and surface temperature at these sites. In all the following sections, overestimated FAC could be due to overestimated snowfall or underestimated meltwater input. Subsurface temperature bias could also be due to biased forcing skin temperature. One obvious pitfall would be to tune firn models so that they can reproduce observations using inaccurate forcing from a RCM. Please discuss this issue and refer to the appropriate literature to validate the forcing used here.

We address your concerns concerning the RCM forcing in the general comments. Again, we agree with you that the climatic forcing brings in a potential bias. But we consider the investigation of this bias as beyond the scope of this study and we assume that the biases related to the liquid water scheme and the firn densification model are greater in amplitude.

p.14 l.7 Here an in the following, a "reasonable" match between modeled and simulated density is often presented after an over- or underestimation of FAC. The two quantity being linked, they should reflect the same conclusions.

We modified the manuscript to ensure that any comparison between different profiles is quantified. In this particular case, the differences in FAC of each of the flow schemes with respect to the observed value are given in the previous lines. A close agreement in FAC values reflect a good agreement with the mean density. Additionally, we refer the reader to the figures of density profiles (Fig. 4a in this case) when we discuss the agreement with the observed density in more details.

p.14 l.31 relative to the original BK run or relative to observations? Check throughout the manuscript that for each of these relative change, the baseline is clearly specified.

We clarified that this is with respect to BK wh02 ip810 on this line. We also made sure that the comparative basis is clear throughout the paper, especially in Sect. 4.3 and 4.4.

p.15 l.33 How does the impermeability criteria relate to the layer thickness? A thick layer can refreeze large amounts of water without being categorized impermeable while minor refreezing in small layers immediately cause saturation and subsequent runoff.

The layer thickness is dependent on the model resolution used. In this study, we attach great importance to using a very fine vertical resolution in order to minimize approximations when solving the Richards Equation and also approximations such as the one you are highlighting. The layers' thickness is determined by the accumulation at every 3h time step modelled by the RCM and we apply merging only for layers less than 0.02 m thick.

It is true that thinner layers require less absolute refreezing to have their density raised to the impermeability threshold. Note however that this is partly counteracted by the lower cold content of a thin layer than a thick layer for a given temperature. Thus, thinner layer also have a lower refreezing capacity.

p.16 l.5 give depth range
We proceeded to the modification suggested.

p.16 l.21 It seems that the grain size parametrization has a great impact on the result although there is no observation of firn grain size or clear constrains on the grain growth modules in firn models. This could be highlighted more. The abstract, for example, could be more specific than "numerous uncertainties surrounding firn micro- and macro-structure...".

We modified the abstract to explicitly mention the importance of grain-size on the water flow scheme. We also modified the Discussion section to highlight the spread in results that can be due to the grain-size formulation. We agree that it is important to

emphasize this because grain metamorphism in firn is poorly constrained by observations and thus subject to important uncertainties.

p.16 l.26 This sentence is unnecessary. The following presentation is sufficient for a result section. In the discussion part however, this spread of outcome for different compaction schemes can be brought up again and an appropriate reference where the schemes are being compared can be given.

We removed this sentence as suggested. We highlight further the need for intercomparison experiments in the Discussion section, such as the one of Lundin et al. (2017).

p.16 l.29 Give bias in the text.
We proceeded to the modification suggested.

p.16 l.31 At this site, valuable observations were made by http://dx.doi.org/10.1038/nclimate2899 http://dx.doi.org/10.5194/tc-9-2163-2015 and doi: 10.1017/aog.2016.2 The output of a firn model forced by HIRHAM RCM is presented here: https://doi.org/10.3389/feart.2016.00110 and another model forced by RACMO is presented here: https://doi.org/10.3389/feart.2017.00003 and https://doi.org/10.5194/tc-11-2507-2017. Please compare forcing and model results.

We address the issue of climatic forcing and further use of observations in our answers to your general comments. Please note that we use here the drilled KAN-U core provided by one of your suggested references (Machguth et al., 2016).
Thank you for highlighting the absence of comparison with previous modelling studies. We now compare the results of this study to the ones of the references you mention (Langen et al., 2017; Steger et al., 2017b) in the Discussion section.

p.16 l.33 Why stopping in 2013?

Drilled cores at KAN-U show that the site turned to bare ice after 2013, thus with almost no remaining firn layer. This study focuses on the simulation of water flow in firn and bare ice conditions are irrelevant for this because of the absence of downward percolation. Therefore, we stop the simulation in 2013.

p.16 l.33 Does that mean that this is ablation area?

The RCM simulated climate indeed indicates that the KAN-U site is turning into ablation area over the recent years. This is further confirmed by drilled cores (see previous answer).

p.17 l.16 Mention the role of runoff in this case.

We proceeded to the modification suggested.

p.18 l.5 It could also increase the structural strength of the firn.

It is true that higher densities cause stronger firn and thus decreased densification rates. However, when the impermeability threshold is increased from 810 to 830 kg m$^{-3}$, the increased strength of the simulated ice layers does not lead to a FAC decrease. Indeed, all other things being equal, a layer at 810 kg m$^{-3}$ in an ip810 simulation will further densify faster than a layer at 830 kg m$^{-3}$ in an ip830 simulation. But this faster densification only occurs because the density is lower (and thus the strength decreased) and the result is that the FAC contribution of that particular layer will always remain higher in the ip810 simulation. On the contrary, the higher load exerted by the denser layer of the ip830 simulation on the firn below will cause faster densification and thus decreased FAC in the ip830 simulation.

p.19 l.18 Please give the observed value.

The observed value is already given in p.19 l.15. If you think it is necessary to repeat this value, we would proceed to this adjustment.

p.21 l.11 Meaning bare ice during the summer or ablation area.

This means bare ice in summer indeed. However, it is not necessarily ablation area because the internal accumulation component of the Surface Mass Balance must be accounted for. In this particular case, it is important because a lot of meltwater refreezes locally in the firn column (see Table 5) and this keeps total accumulation above ablation.

p.21 l.15 Why is runoff combined with refreezing here? A more interesting comparison would be solid input + refreezing (mass gain) compared to runoff (mass loss). This would be another formulation of the surface mass balance.

Here we only use this additional statement to provide an explanation to the reader of why the sinks of liquid water (refreezing and runoff) exceed the sources of liquid water (surface melt and rain), which violates mass conservation if one looks only at a single year. We do not combine runoff and refreezing for assessing any other variable and we agree with your statement of Surface Mass Balance quantification. However, we think that the manuscript is already long and so we do not provide this detailed quantification of annual Surface Mass Balance.

p.21 l.15 Maybe split this sentence.
We proceeded to the modification suggested.

p.21 l.19 The difference between R1M and DPM as well as the sensitivity of DPM to its internal parameters was discussed in Wever et al. 2016. Please compare your results to their work on alpine snowpack. Do you see the same patterns? Did they see for example the sensitivity of DPM to the grain size formulation?

In their study, Wever et al. (2016) also showed the greater ability of DPM to produce ice layers than R1M. We added this valuable comparison in the Discussion section. However, Wever et al. (2016) only tested the sensitivity of their model to their tuning parameters $\Theta$ and $N$, that we keep constant in this study (see Sect. 3.3 and Sect. 4). They did not try different grain-size formulations and it is thus impossible to know if they have noticed an important sensitivity of the flow schemes to grain-size, as we do in this study.

p.21 l.20 Here "greater retention" seem to contradict with favored lateral runoff. Please clarify.

Indeed the term "greater retention" is confusing and inappropriate. We replaced it by "slower downward percolation".

p.21 l.25 I believe "biases" cannot be hot or cold. Please rephrase.

We proceeded to a rephrasing as demanded.

p.21 l.28 This is quite surprising when the BK wh02 was made to mimic piping processes and R1M is supposed to ignore it. Please discuss.

We modified Sect. 5 in the manuscript to discuss more the agreement between R1M and BK and what this implies for situations in which BK is tuned to mimic preferential flow.

p.22 l.23 Due to the impermeability threshold, the ice density in the model is very low (max. 830 kg/m3 vs. ~900kg/m3 at KAN_U and vs. 917kg/m3 in the FAC calculation). This skews completely the comparison of FAC. Here DPM gives the same FAC as the firn cores in spite of simulating completely saturated firn below 2.5 m.

We agree with your comment and so have removed this section from the revised version of the manuscript.

p.23 l.13 Can your experiment tell anything about what should be developed/improved in the DPM ?

We think indeed that our study provides some insights of which aspects of DPM should be improved. We modified the manuscript to highlight these more clearly by summarizing the three main shortfalls of DPM. Note that these modifications are made further in the Discussion section:
*We highlight (1) that too much water is transported through preferential flow or at exaggerated depths as demonstrated by the consistent overestimation of 10 m temperature, (2) the need to consider lateral flow as the tendency to underestimate FAC shows and (3) the large sensitivity of the ice layer formation process to grain size, which should be overcome or should be addressed by further observational studies of grain metamorphism in firn.*

p.23 l.14 I do not understand why the presented biases may be related to a sensitivity of the model to temperature. Could you specify?

We argue that the densification models are not well suited for conditions where the refreezing of meltwater causes latent heat release and thus higher firn temperatures. We think that these high temperatures cause overestimated densification rates as has been pointed out in previous studies (e.g. Steger et al., 2017a). We modified the manuscript to clarify this point.

p.23 l.29 Maybe "water flow" ?
We proceeded to the modification suggested.

p.24 l.21 Be more specific here, there has been work in Greenland and elsewhere about the spatial heterogeneity in firn (and in FAC). One source could be doi: 10.1017/jog.2016.118

We modified the manuscript to emphasize that observational constrains of firn structure are affected by strong spatial variability and we cite the reference you suggested.

p.25 l.23 Rephrase. It deteriorates the performance of the model or "explains the mismatch". But the validation (the act of validating) in itself is always possible.

The validation is affected by the fact that the liquid water scheme is far from being the only component of the firn model to influence depth-density profiles. Thus there are many sources of uncertainty at play and the problem of compensating errors can

arise. Continuous measurements of liquid water content would provide a more direct way to validate liquid water schemes, but such measurements are not available. We modified the manuscript to clarify this point.

p.25 l.25 Be specific. You have identified some crucial but unconstrained parameters such as grain size. Or are you referring to processes not yet explored? The second half of this sentence is also very redundant with the firt part.

We modified the manuscript to identify more clearly the poorly constrained complexities affecting the flow scheme. We split the sentence in two separate sentences. The second part of the statement is meant to identify the processes not immediately related to the flow scheme that must be better represented to reach more accurate modelled firn densities in the percolation area. We hope that the rephrasing helps clarify all of this.

p.26 l.6 Please give websites for PROMICE and DOI for SUMup.

We proceeded to the modifications requested.

p.27 l.12 1 m w.eq. I guess?

The variable is the volumetric water content. Thus it is the ratio between the amount of water (in m w.e.) in a layer and the thickness of the layer (regardless of the density of the layer).

p.27 l.17 It seems that this minimum liquid water for refreezing is supposed to solve the same issue (necessary residual water in all layers) as eq. 19. How do these two steps work together?

Indeed, this 0.01 % value and the adjustment of $\theta_r$ have the same numerical purpose: the effective saturation of any layer cannot reach zero. However, they influence different processes: $\theta_r$ influences flow rates and the 0.01 % value influences refreezing rates. The restriction on refreezing rates comes into play only once $\theta$ falls below $10^{-4}$ (0.01 %), whereas the adjustment of $\theta_r$ is required as soon as $\theta$ approaches 0.02. Please note that the amounts of refreezing that are omitted due to this 0.01 % value would be a negligible fraction of the total refreezing in any of our simulation but is only required for numerical purposes.

p.27 l.18 Is time step meant here? Or step in the iterative solving of water movement?

Yes, time step is meant here. At any point in the solving of the RE, a layer cannot be completely dry. The artificial initialisation of all layers at $\theta_{dry}$ is executed only at the beginning of the flow routine and not at every iterative step. Therefore, the complete refreezing can only be applied at the end of the flow routine and not at the end of every iterative step. The latter case would lead to layers being dry when the next iterative step starts.

p.27 l.20 Here the convergence criteria is also used as a liquid water threshold for dry conditions. Maybe you could give different names to these variables even though they have the same value.

We replaced the use of $\varepsilon_\theta$ by its numerical value $10^{-5}$ as suggested (it is used only once, so we preferred not to define a new variable name).

**Referee 3**

We thank this reviewer for their compliments and feedback. Our response to these comments is below, the referees original text is given in red throughout and our response is in black.

General comments

1) Measured temperature was used for the validation of models. Simulated runoff was also discussed. However, figures for these results were not shown. Lack of figures sometimes makes difficult to understand in detail. In TC, authors can use a supplement file to show figures relatively less important. Therefore, figures of them should be added using a supplement to support the discussion in the main text (see minor comment P13 L22-28, P14 L14-15).

We thank you for this constructive suggestion. We have added in a Supplementary Information file the figures of the evolution of annual runoff and annual refreezing, showed with the annual fluxes of meltwater as a basis of comparison. This can indeed help the reader to have a better overview of the evolution of these variables in time. We also added figures of the evolution of the modelled temperature profiles. This is especially valuable for highlighting how the use of different flow schemes affect the thermal properties of the modelled firn layer. As an example, it is easy for the reader to see how DPM tends to bring too much heat at great depths using figures S5 and S6. Please note that we did not add figures showing the differences between modelled temperature profiles and measured temperature profiles in time. Firstly, the evolution of the temperature in depth has not been measured systematically at all the sites tested. Also, our point of view, supported by Referee 1 and Referee 2, is that we should not expect a model configuration to reproduce precisely a point measurement for the following reasons: (i) the climatic forcing comes from a Regional Climate Model of 5.5 km horizontal resolution, (ii) firn properties show large spatial variability. We added more clarification about argument (ii) in the introduction of Sect. 4 and in Sect. 5 in the revised version of the manuscript. We use a comparison with the 10 m depth firn temperature because we think that this gives a good evaluation of the ability of the different models to reproduce the bulk thermal properties of the firn layer. However, we think that showing a detailed difference of the modelled and measured temperature in depth and in time may mislead the reader to think that the model is expected to reproduce observed values with great precision despite (i), (ii) and the many sources of additional uncertainty that we highlight in the manuscript.

Section 4: *Comparing the modelled FAC and 10 m depth temperature values with observed data depicts the ability of the tested models to reproduce the bulk condition of the upper firn column. We also qualitatively assess the degree to which the models to form a 'realistic' ice layer distribution and depth-density profile. One would not expect simulated values of either to match observations precisely given the high spatial variability of firn structure (Marchenko et al., 2017), but it is indicative of the models' performance in reproducing heterogeneity in firn density.*

Section 5: *On the other hand, comparisons between modelled and observed density profiles are strongly affected by the choice and accuracy of the densification formulation, the variability of surface density, several other factors influencing model outputs mentioned above and possible uncertainties in field measurements. Such uncertainties are related to the strong spatial variability of firn structure (Marchenko et al., 2017), which can be observed by comparing density profiles of cores drilled at nearby locations.*

2) The simulation in this paper was performed for four fields. Tables were provided for results in each field showing total or averaged simulated values. I would like to suggest that the author provide a table which shows the comprehensive result to see the difference between fields about simulation results (see minor comment in P19L5 and P22L26). This is not prerequisite for acceptance, but it will help to understand the overall result.

We have also added in the Supplementary Information file a table that summarizes the difference between measured and modelled values of each flow scheme at every site for the FAC15 and the 10 m temperature. We did not put the table in the main text of the manuscript because most of the values are already given in the Tables 2, 3, 4 and 5. We hope that this fulfils your request of a comprehensive table.

3) Although the main target of this study was the validation of liquid water infiltration model, density and temperature data were used instead of liquid water content for validation. It leads to the limitation of the validation itself. Discussion about the limitation because of this is also necessary.

The limitations of the validation process for firn percolation schemes is one of the challenges that our study highlights. Examples of such limitations are the unavailability of liquid water flow measurements in firn and the competing effects of other processes on the density profile (compaction, surface density, etc.). We modified the conclusion section in the revised manuscript in order to ensure this was adequately addressed:

*There is no large scale detailed observation available of liquid water content and percolation pattern during melting events in firn. This renders the validation of a particular flow scheme difficult and validation relies on temperature and density profiles. However, there are a number of effects that influence firn density and temperature, all potentially contributing to mismatch between modelled and observed values.*

4) This simulation study can provide several suggestions for a laboratory experiment and field observation required to improve the model. Although limitations are written in conclusion, detailed discussion about limitation and suggestion of new experiment and observation (e.g. liquid water infiltration experiment into firn or observation of liquid water.) will be informative for future research.

Our study tries to outline the limitations of current liquid water flow scheme when applied within firn models. We have additionally modified the revised version of the manuscript in order to provide a detailed summary of the shortcomings of DPM in the Discussion section. We hope that our findings could help laboratory and field researchers to design experiments that would address the numerous uncertainties highlighted throughout the article. We leave it up to these specialists to identify the experiments that would best address these limitations as this is beyond our area of expertise.

Section 5: *We highlight (1) that too much water is transported through preferential flow or at exaggerated depths as demonstrated by the consistent overestimation of 10 m temperature, (2) the need to consider lateral flow as the tendency to underestimate FAC shows and (3) the large sensitivity of the ice layer formation process to grain size, which should be overcome or should be addressed by further observational studies of grain metamorphism in firn.*

Specific comments

P8 L26-27 In this sentence and Eq. (17), the saturated water content was estimated as i/w (0.917?) of pore space. It seems to be used for convenience in calculation. Yamaguchi et al. (2010) obtained the saturated water content was about 90% of pore space in their gravity drainage column experiment. Although they are coincidence, this paper had better be referred to show that the assumption in Eq. (17) is consistent.

Our approach in this study has been to implement the DPM flow scheme following the SNOWPACK dual permeability water scheme with the greatest fidelity possible. Because the SNOWPACK scheme uses that correction for the saturation, we have decided to implement the same. Indeed, this correction for saturated water content is in good agreement with the findings of Yamaguchi et al. (2010). We mention this in the revised form of the manuscript as you suggested and we also refer to the laboratory study of Yamaguchi et al. (2010).

P13 L22-28 Figures for simulated runoff had better to be shown in a supplement.

As we explained in the answer to your major comment 1), these figures have been added to a Supplementary Information file.

P14 L14-15 Comparison between observation and simulation of temperature also needs a figure in a supplement.

This point has also been addressed in the answer to your major comment 1).

P17 L18-30 In terms of average value, DPM had a good agreement. However, the simulated result of DPM was a constant value for vertical and maximum density was underestimated. In my opinion, the depiction "good agreement" feels not suitable (depiction "average density is reasonable" may be OK). The state of this result had better be that present liquid water infiltration scheme has limitation and requires future improvement.

We proceeded to the suggested modification.

P19L5 Here, densification model had a larger effect than water infiltration model. Is this trend only this place or common for all four places? Comprehensive table comparing between fields about this will help to check this question. (This is suggested in major comments.)

We have also added in the Supplementary Information a table that summarizes the difference between measured and modelled values of each densification scheme at every site for the FAC15 and the 10 m temperature. By comparing the values of both tables that have been added in the Supplementary Information, the reader can have a good overview of the respective effect of the choice of the water schemes and of the choice of the densification model.

Please note also that we address the great variability at all sites due to the densification scheme used in the Discussion and Conclusion sections. To emphasize this more, we added in the Discussion section a mention to the need for more firn models intercomparison experiments.

P20 L27 L31 Fig8f? Recheck the figure number
This was a typo in the first version of the paper. Thank you for pointing this out.

P22 L7-9 Is it mean that the simulation using BK and R1M were performed by CROCUS whereas DPM was performed by SNOWPACK? If so, the difference in results receives the effect of the difference of numerical snowpack model. Did authors check the difference of them performing a simulation with the same water infiltration scheme? SNOWPACK has Bucket and RE scheme comparable with BK and R1M, respectively.

This is not the case. All water schemes were always used with the same densification model (CROCUS for the sites DYE-2, NASA-SE and KAN-U, HL for the FA13 site as explained in the text). This was indeed to avoid differences in results to arise from the use of different firn densification models. The DPM water scheme implemented in the firn model is a reproduction of the dual permeability water scheme that was implemented in SNOWPACK by Wever et al. (2016). What we explain in this paragraph is that a simulation performed by the full SNOWPACK model would be different from our simulations because SNOWPACK has different physics for densification and grain growth for example (which have not been implemented in the firn model). We modified the revised version of the manuscript in order to clarify this.

P22 L26 This results seems that the DPM is not suitable for this place (actually, improvement is necessary for preferential flow scheme). As discussed in P17 L4-8, BK and R1M reproduce surface ponding and refreeze. If a suitable model is different depending on the field, the most suitable scheme had better be shown for each field. Comprehensive table suggested in a major comment is useful to show a comparison between fields.

This results suggests indeed that the current form of DPM is not suitable for the conditions at KAN-U. We attached great importance throughout the entire study to show the performances of each of the flow schemes (BK, R1M, DPM) at the four different sites and not to systematically pick the water scheme that gave results in closest agreement with observations. This is because the aim of this study is to give a first overview of the performance of physically based water schemes of snow models (R1M and DPM) when applied in firn models and to compare these with the common bucket scheme (BK). We do not argue about whether a particular model is better than the other and we try to provide some ways of improvements for further development of water flow schemes in firn models. By highlighting the deficiencies of DPM, we also hope to orientate future modelling efforts. We hope that the tables added in the Supplementary Information satisfy your requests of comprehensive tables. Please, note also that this specific part of the Discussion section has been removed from the revised version of the manuscript. In accordance with the point of view of other referees, we deemed it too speculative.

[revised manuscript text omitted]

**Figures**

[Figure]

**Figure S1. Annual rates of liquid water input, runoff and refreezing at DYE-2 as simulated by (a) BK wh02 ip810, (b) R1M grLK ip810 and (c) DPM grLK ip810**

[Figure]

**Figure S2. Annual rates of liquid water input, runoff and refreezing at NASA-SE as simulated by (a) BK wh02 ip810, (b) R1M grLK ip810 and (c) DPM grLK ip810**

[Figure]

**Figure S3. Annual rates of liquid water input, runoff and refreezing at KAN-U as simulated by (a) BK wh02 ip810, (b) R1M grLK ip810 and (c) DPM grLK ip810**

[Figure]

**Figure S4.** Annual rates of liquid water input, runoff, refreezing and storage of liquid water in the firn column at FA13 as simulated by (a) BK wh02 ip810, (b) R1M grLK ip810 and (c) DPM grLK ip810

[Figure]

**Figure S5. Modelled firn temperature at DYE-2, (a) BK wh02 ip810, (b) R1M grLK ip810, (c) DPM grLK ip810, (d) BK whCL ip810, (e) R1M grA ip810, (f) DPM grA ip810**

[Figure]

**Figure S6. Modelled firn temperature at KAN-U, (a) BK wh02 ip810, (b) R1M grLK ip810, (c) DPM grLK ip810, (d) BK whCL ip810, (e) R1M grA ip810, (f) DPM grA ip810**

[Figure]

**Figure S7.** Modelled firn temperature at FA13, (a) BK wh02 ip810, (b) R1M grLK ip810, (c) DPM grLK ip810, (d) BK whCL ip810, (e) R1M grA ip810, (f) DPM grA ip810

---

## Author Response (AR2)

Dear Nanna Karlsson,

We thank you for your comments on the manuscript. We addressed the two outstanding issues that you have highlighted. We justify further our selection of firn cores in Section 2.4 and we based our modifications on your suggestions. We discuss more thoroughly the possible bias introduced by the climatic forcing. We introduce this subject in Section 2.2 and we refer to the Supplementary Material. In the Supplementary Material, we added an entire section in which we compare RACMO output with Automatic Weather Station data at our study sites, and the discussion is supported by relevant graphs and a Table of statistics of the comparison. Additionally, at the end of the Discussion section in the main manuscript, we give more details on the effect of climatic forcing errors on the simulations of the firn models. Concerning your minor comments, we proceeded to the modifications suggested.

We hope you will be satisfied with our edits.

Best regards,

Vincent Verjans, on behalf of the authors